Letter

# A sequence of SVA retrotransposon insertions in *ASIP* shaped human pigmentation

Nolan Kamitaki [1,2,3,4,5,6] ✉, Margaux L. A. Hujoel [1,2,3], Ronen E. Mukamel[1,2,3], Edward Gebara [3,4,5], Steven A. McCarroll [3,4,5] ✉ & Po-Ru Loh [1,2,3] ✉

Retrotransposons comprise about 45% of the human genome[1], but their contributions to human trait variation and evolution are only beginning to be explored[2,3]. Here, we find that a sequence of SVA retrotransposon insertions in an early intron of the *ASIP* (agouti signaling protein) gene has probably shaped human pigmentation several times. In the UK Biobank (*n* = 169,641), a recent 3.3-kb SVA insertion polymorphism associated strongly with lighter skin pigmentation (0.22 [0.21–0.23] s.d.; $P = 2.8 \times 10^{-351}$) and increased skin cancer risk (odds ratio = 1.23 [1.18–1.27]; $P = 1.3 \times 10^{-28}$), appearing to underlie one of the strongest common genetic influences on these phenotypes within European populations[4–6]. *ASIP* expression in skin displayed the same association pattern, with the SVA insertion allele exhibiting 2.2-fold (1.9–2.6) increased expression. This effect had an unusual apparent mechanism: an earlier, nonpolymorphic, human-specific SVA retrotransposon 3.9 kb upstream appeared to have caused *ASIP* hypofunction by nonproductive splicing, which the new (polymorphic) SVA insertion largely eliminated. Extended haplotype homozygosity indicated that the insertion allele has risen to allele frequencies up to 11% in European populations over the past several thousand years. These results indicate that a sequence of retrotransposon insertions contributed to a species-wide increase, then a local decrease, of human pigmentation.

Variation in skin pigmentation has profoundly influenced human evolution and social history, enabling *Homo sapiens* to adapt to environments with diverse levels of solar radiation. Agouti signaling protein (ASIP) is a secreted protein that plays a key role in skin and hair pigmentation by binding to a receptor (melanocortin 1 receptor (MC1R)) on the surface of melanocytes, causing them to shift melanin pigment production from darker, brown eumelanin to lighter, red pheomelanin[7]. Across vertebrates, regulated increases in expression of ASIP decrease

pigmentation temporally and in different parts of the body[8]. In humans, the ASIP-MC1R pathway is affected by several of the largest influences of common genetic variation on skin and hair pigmentation, including several common missense variants in *MC1R*[9,10].

Genome-wide association studies (GWAS) in European genetic-ancestry cohorts for pigmentation-related traits, including skin cancers such as melanoma, have long observed a particularly strong association near the *ASIP* gene[4–6] that colocalizes with an expression

[1]Division of Genetics, Department of Medicine, Brigham and Women's Hospital and Harvard Medical School, Boston, MA, USA. [2]Center for Data Sciences, Brigham and Women's Hospital, Boston, MA, USA. [3]Program in Medical and Population Genetics, Broad Institute of MIT and Harvard, Cambridge, MA, USA. [4]Stanley Center for Psychiatric Research, Broad Institute of MIT and Harvard, Cambridge, MA, USA. [5]Department of Genetics, Harvard Medical School, Boston, MA, USA. [6]Department of Biomedical Informatics, Harvard Medical School, Boston, MA, USA. ✉e-mail: nolan_kamitaki@hms.harvard.edu; smccarro@broadinstitute.org; poruloh@broadinstitute.org

**Fig. 1 | Characterization of a polymorphic SVA F₁ retrotransposon insertion within an intron of *ASIP*. a**, Architecture of *ASIP* isoforms and composition of the SVA F₁ retrotransposon insertion. *ASIP* has four known alternate first exons (dark blue), with the first three (exons 1A, 1B and 1C) able to either splice directly to a coding exon (gold; exon 3) or first to an additional 5′ UTR exon (light blue; exon 2). The SVA F₁ insertion contains the expected 5′ truncation with *MAST2* exon followed by VNTR and SINE-R sequences. The location of the SVA F₁ insertion, present in the GRCh38 reference, is indicated by the purple thin vertical bar. **b**, Pairwise sequence alignment dotplot of long-read sequencing-derived haplotypes[23] from an individual heterozygous for the SVA F₁ insertion (NA12329). The SVA F₁ insertion, present in the haplotype on the *y* axis, corresponds to the vertical break in the diagonal alignment (purple) and bears substantial homology (dashes to the left of the vertical break) to a nonpolymorphic SVA F retrotransposon 3.9 kb upstream (light purple). **c**, Genotyping approach for short-read data. Read alignments overlapping the right breakpoint of the SVA F₁ indicate the presence of at least one insertion allele, whereas discordant read pairs with excessively long fragment sizes indicate the presence of at least one non-insertion allele (Methods). CDS, coding sequence. **d**, Determination of SVA insertion genotypes for individuals in the 1KGP. Individuals homozygous for the ancestral allele without the SVA F₁ insertion (Hom-ANC; gray) are the most common and have no or few reads overlapping the right breakpoint; individuals homozygous for the insertion allele (Hom-INS; purple) are the least frequent and have few or no reads with fragment size >2.5 kb; heterozygous individuals have both read types (Het; pink). **e**, SVA F₁ insertion allele frequency across 1KGP populations. The insertion is present in European genetic ancestry populations and others with known European admixture but is otherwise absent. **f**, Extensive LD (*r²*) of the SVA F₁ insertion (at 34.2 Mb) with variants in a 5-Mb window on chromosome (chr) 20 (32.5–37.5 Mb; GRCh38 coordinates) in CEU and GBR 1KGP populations.

quantitative trait locus (eQTL) for *ASIP*[11]. However, a plausible functional variant for this common, large effect (>0.2 s.d. change in pigmentation phenotypes) has not been identified, despite the considerable statistical resolution afforded by large biobank cohorts[12]. GWAS of African cohorts (from Ethiopia, Tanzania, Botswana or KhoeSan populations)[13,14] or East Asian cohorts (from Japan or Korea)[15,16] have not found an association at this locus, suggesting that the functional variant emerged recently and on a European-specific haplotype, consistent with a genome-wide scan of recent positive selection in a British cohort that identified the *ASIP* locus among other pigmentation-associated genes[17].

Across mammals, structural mutation at *ASIP* is a recurring mechanism underlying variation in coat color. Changes in coat color have occurred by large rearrangements at *ASIP* in lethal yellow agouti mice (*Aʸ*)[18], sheep[19] and gibbons[20], and by retrotransposon insertions in

viable yellow agouti mice (*Aᵛʸ*)[21] and dogs[22]. However, such polymorphisms have not been reported for human *ASIP*.

To identify structural variation at *ASIP* that could underlie genetic associations with pigmentation, we examined long-read sequence assemblies generated by the Human Genome Structural Variation Consortium[23] (HGSVC2; *n* = 32). The single individual that was heterozygous for the light-pigmentation/cancer-risk haplotype (NA12329) was also heterozygous for a large, 3.3-kb structural variant in an intron of *ASIP* (Fig. 1a,b) previously inferred from short-read sequencing analyses[24,25]. Optical mapping data (Bionano) confirmed this insertion as the only large structural variant within 500 kb of *ASIP* carried by NA12329. The variant overlaps a SINE-VNTR-Alu (SVA) element annotated by Repeat-Masker[26] (in antisense orientation relative to *ASIP* transcription) at chr20:34228123–34231419 in the GRCh38 reference, suggesting that

the human genome (GRCh38) reference sequence has an allele with a recent, polymorphic SVA insertion that is in fact absent from most *ASIP* haplotypes. SVA elements are an active, recent family of retrotransposons unique to great apes, with the E, F and $F_1$ subfamilies specific to humans[27]. Sequence evidence suggests that this SVA is in the youngest SVA $F_1$ subfamily[28,29], as it lacks a key 5′ (CCCTCT)n hexameric repeat and Alu-like elements and also contains 92 bp matching the *MAST2* exon 1 that was 5′-transduced into the subfamily's source element. Notably, this polymorphic SVA is 3.9 kb downstream of (and in the opposite orientation to) another, nonpolymorphic 1.6-kb SVA F retrotransposon within the same intron of *ASIP* (Fig. 1b).

To facilitate deeper analysis of this variant in larger, phenotyped cohorts, we devised a strategy for ascertaining individual-level genetic states (genotypes) from short-read whole-genome sequencing (WGS) alignment patterns specific to each allele (Fig. 1c; Methods). Applying this approach to high-coverage WGS of 1000 Genomes Project (1KGP) samples[30] demonstrated good separation of genotype clusters (Fig. 1d). Across 1KGP population samples, the SVA $F_1$ insertion exhibited the highest allele frequencies (7–8%) in the northwest European (GBR and CEU) population samples and was not detected in (nonadmixed) samples of a variety of populations from Africa and Asia (Fig. 1e). In CEU and GBR population samples, the SVA $F_1$ insertion was in strong linkage disequilibrium (LD) ($r^2 = 0.93$) with the pigmentation-associated index single nucleotide polymorphism (SNP) rs6059655, and LD with other SNPs spanned a ~5-Mb extended haplotype (Fig. 1f).

We applied the same SVA genotyping approach to WGS data available for 169,641 unrelated White individuals of European genetic ancestry in the deeply phenotyped UK Biobank (UKB) cohort[31,32] (Fig. 2a), finding the allele frequency of the insertion to be 11%. We estimated the accuracy of genotyping to be $r^2 \approx 0.997$ based on the concordance of genotype calls across sibling pairs sharing both *ASIP* alleles identically by descent (IBD2) (Fig. 2b). Across pigmentation traits including skin color, hair color and tanning response, the SVA $F_1$ insertion associated more strongly with lighter pigmentation (0.22 [95% confidence interval (CI):0.21–0.23], 0.24 [0.23–0.25] and 0.27 [0.26–0.28] s.d.; $P = 2.8 \times 10^{-351}$, $2.0 \times 10^{-396}$ and $1.5 \times 10^{-523}$, respectively) than did all other SNP and indel variants in the region, explaining 0.9–1.4% of trait variance (Fig. 2c,d and Extended Data Figs. 1a,b and 2a). Likewise, the SVA $F_1$ insertion associated more strongly with increased skin cancer risk (odds ratio (OR) = 1.23 [1.18–1.27]; $P = 1.3 \times 10^{-28}$) than did any nearby variant (Fig. 2e,f and Extended Data Figs. 1c,d and 2b). In a joint analysis (of both the SVA $F_1$ insertion and the lead SNP rs6059655) for tanning response (the pigmentation trait with the strongest association at the locus), the SVA $F_1$ insertion remained significantly associated ($P = 6.9 \times 10^{-31}$), whereas the lead SNP did not ($P = 0.56$). Fine-mapping analysis using SuSiE[33] also selected the SVA as the only member of a single credible set. Conditional association analyses including the SVA as a covariate further suggested that the SVA $F_1$ insertion almost completely accounted for pigmentation associations at the locus: residual signal was only 1–2% as strong (Extended Data Fig. 1e–j).

As SNPs on this haplotype have been observed to associate with *ASIP* expression levels in skin[11], we next asked whether this insertion is the likely cause of this effect on *ASIP* expression. Genotyping the SVA $F_1$ insertion in WGS data available for tissue donors of the Genotype-Tissue Expression (GTEx) Project[34] (Extended Data Fig. 3) showed that, as with the pigmentation associations, the insertion associated strongly with *ASIP* expression in both sun-exposed (SE) and not sun-exposed (NSE) skin ($P = 3.5 \times 10^{-17}$ and $1.3 \times 10^{-21}$, respectively), and that the insertion appeared to account for most of the eQTL signal at the locus (Fig. 3a–d and Extended Data Fig. 4). The SVA insertion associated with a 2.2-fold (1.9–2.6) increase in *ASIP* expression in NSE skin. Closer examination of RNA sequencing (RNA-seq) read alignments at *ASIP* showed substantial RNA-seq coverage at several alternative first exons as well as within introns (Fig. 3e). Whereas the SVA $F_1$ insertion associated with broadly increased expression across all exons, it associated with decreased

abundance of unspliced transcripts containing intronic sequence upstream of the SVA (Fig. 3f).

We therefore hypothesized that the SVA $F_1$ insertion increases *ASIP* expression by improving splicing fidelity (and thus reducing the ascertainment of unspliced transcripts). To test this idea, we analyzed all the *ASIP* splice junctions observed in GTEx skin samples (reported by LeafCutter[35]). One of the more frequent anomalous splice events involved splicing from an *ASIP* 5′ untranslated region (UTR) exon to a computationally predicted splice acceptor (SpliceAI[36] acceptor probability of 0.51) within the nonpolymorphic SVA F element that resides in the same intron (in opposite orientation) as the polymorphic SVA $F_1$ insertion (Fig. 4a,b). In heterozygotes for the SVA $F_1$ insertion, the relative frequency of splicing into the SVA F element, rather than the downstream coding exon, decreased from 14.1% to 3.1% in SE skin and 15.6% to 5.6% in NSE skin ($P = 1.6 \times 10^{-21}$ and $1.0 \times 10^{-11}$, respectively; Fig. 4c,d). In three GTEx donors homozygous for the SVA $F_1$ insertion, no evidence of splicing into the SVA F element was observed. As with the expression QTL, the SVA $F_1$ insertion seemed to explain this splicing QTL signal (Extended Data Fig. 5). Scanning downstream to determine the fate of these aberrant transcripts revealed a termination point of the intronic read alignments, at which several reads ended in poly(A) sequences (Fig. 4a,b), concordant with a polyadenylation site predicted by APARENT[37] (Fig. 4b). These observations together indicate that the aberrant transcripts spliced into the upstream SVA F element are terminated to yield a noncoding transcript, and that the presence of the polymorphic SVA $F_1$ insertion inhibits the production of such transcripts while increasing the production of ASIP-coding transcripts. Because translation stabilizes transcripts, this analysis may underestimate the relative amount of noncoding transcript being produced, as suggested by the fold increase in expression. We propose that the polymorphic SVA $F_1$ element—inserted in inverse orientation relative to the upstream SVA F element—forms (together with the upstream SVA F) a hairpin structure that blocks the function of splice enhancers and/or splice acceptor sequences in the SVA F element, ensuring productive splicing to ASIP-coding exons downstream of the SVA insertions (Fig. 4e). This model resembles recent reports of inverted pairs of Alu elements modulating splicing via formation of an RNA hairpin[3,38].

These results led us to hypothesize that ancient *ASIP* alleles that predated both SVA insertions spliced *ASIP* more similarly to the splicing yielded by the present-day derived haplotype that contains both SVAs. Human genetic data do not enable assessment of this question because the upstream SVA F element, which is human-specific (Extended Data Fig. 6a), has reached fixation in present-day human populations (Extended Data Fig. 6b). We therefore used an in vitro construct to verify that the SVA F element can function as a splicing acceptor when inserted in the hybrid intron of the CAG promoter[39] (Extended Data Fig. 7a; Methods), similar to SVA splicing constructs from previous work[26]. Insertion of the SVA F element caused approximately 11% of transcripts to splice into the SVA at the same aberrant acceptor site as in *ASIP* (Extended Data Fig. 7b–d), consistent with the idea that the ancient human (no longer polymorphic) SVA insertion had reduced productive splicing of *ASIP*.

In contrast to the nonpolymorphic SVA F element, the polymorphic SVA $F_1$ insertion allele (the reference, minor allele) was detected only in population samples with European genetic ancestry. Furthermore, this ASIP-expression-increasing, pigmentation-decreasing SVA $F_1$ insertion allele exhibited long-range (>3 Mb) LD—generally a property of recent mutations—on European haplotypes (Fig. 1f and Fig. 5a,b), whereas such long-range LD was not observed in non-European population samples (Fig. 5c). Analysis of haplotype genealogies in 1KGP CEU and GBR population samples (Methods) dated the insertion at 16,400–21,800 years ago and 14,300–25,400 years ago, respectively (Extended Data Fig. 8). These lines of evidence suggest that this SVA insertion has increased quickly in allele frequency relative to other

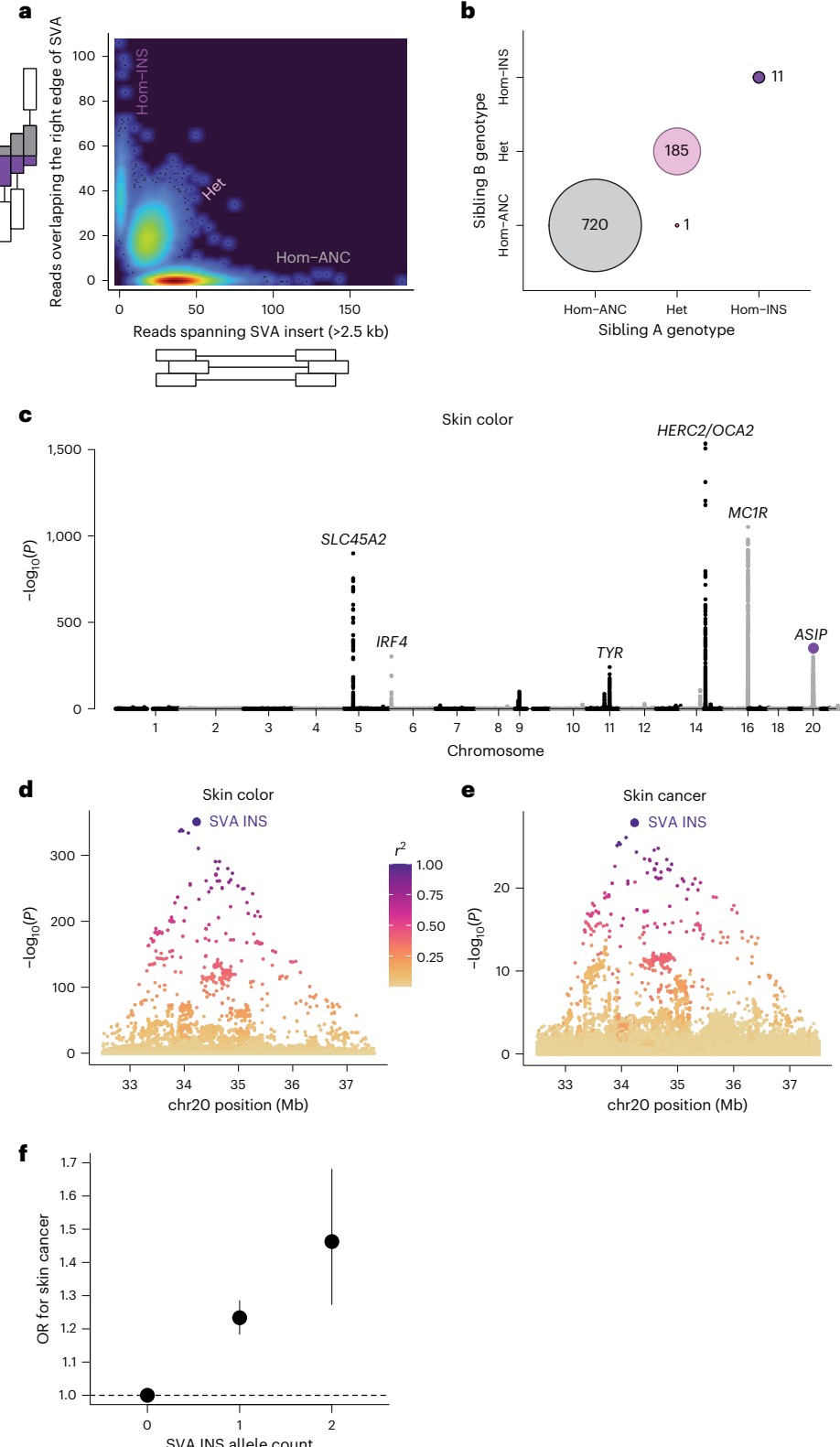

**Fig. 2 | Association of SVA $F_1$ insertion with pigmentation phenotypes in UKB. a**, Genotyping of 199,956 UKB participants with WGS. Three genotype clusters corresponding to the dosage of the SVA $F_1$ insertion are apparent. **b**, Genotype concordance of 917 sibling pairs sharing both *ASIP* haplotypes IBD2. All but one of the sibling pairs agree on the genotype call made for the SVA $F_1$ insertion. **c**, Associations of genome-wide imputed SNP and indel variants with self-reported skin color, coded on a scale from fairest to darkest ($n = 167,568$); $P$ values are from linear regression. The association of the SVA $F_1$ insertion is plotted in purple. **d**, Local association plot for skin color

($n = 167,568$) at the extended *ASIP* locus. Association strengths track with LD with the SVA $F_1$ insertion (yellow-to-purple shading), indicated by the large purple dot. **e**, Associations at *ASIP* with any type of skin cancer (C43 or C44 ICD-10 code; $n_{controls} = 154,340$ and $n_{cases} = 15,295$). **f**, ORs for skin cancer risk for individuals heterozygous ($n = 31,932$) or homozygous ($n = 1,952$) for the SVA $F_1$ insertion relative to individuals homozygous for no insertion ($n = 135,751$). Centers, effect size estimate for each genotype from logistic regression; error bars, 95% CIs.

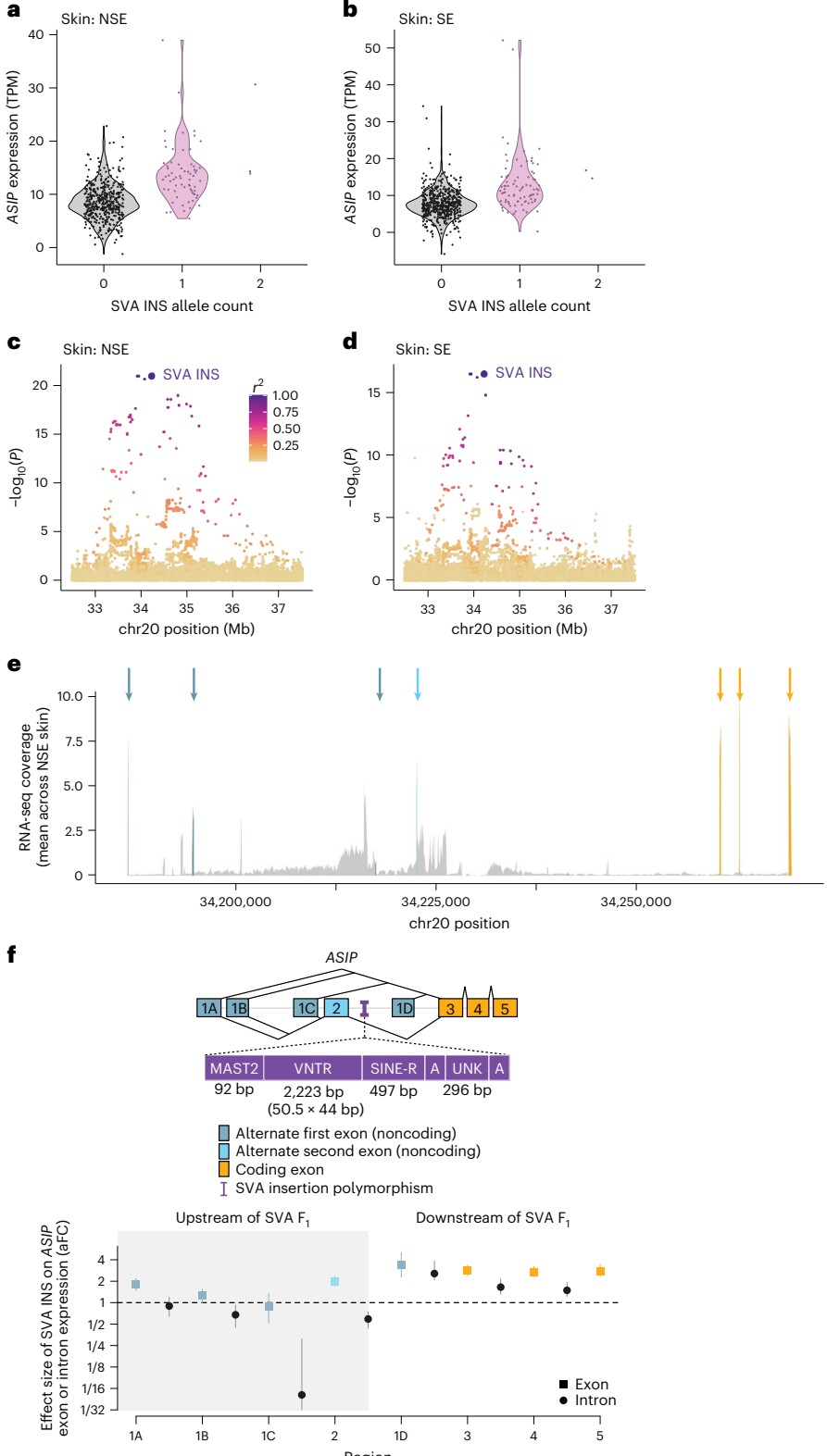

**Fig. 3 | Association of SVA F$_1$ insertion with expression of *ASIP* exons and introns in skin. a**, *ASIP* gene expression in GTEx NSE skin samples ($n = 517$) stratified by SVA F$_1$ insertion genotype. **b**, Analogous to **a**, for GTEx SE skin samples ($n = 605$). **c**, Local association plot for *ASIP* gene expression in NSE skin samples ($n = 517$). Association strengths track with LD with the SVA F$_1$ insertion (yellow-to-purple shading). **d**, Analogous to **c**, for GTEx SE skin samples ($n = 605$). **e**, Depth of coverage of RNA-seq read alignments at *ASIP*, averaged across NSE skin samples. Coverage at exons is indicated with colors corresponding to the *ASIP* gene model below (dark blue, alternative first 5′ UTR exons; light blue, optional second 5′ UTR exon; gold, coding exons). RNA-seq coverage in introns is indicated in gray. **f**, Effect size of SVA F$_1$ insertion for expression of each *ASIP* exon and intron (in units of allelic fold change; aFC) in NSE skin samples ($n = 517$). Intronic regions are defined between adjacent exons; measurements in these regions presumably correspond to prespliced mRNA, potentially from several isoforms. Centers, point estimate of aFC from linear regression coefficients for the indicated region; error bars, 95% CIs from bias-corrected and accelerated bootstrap.

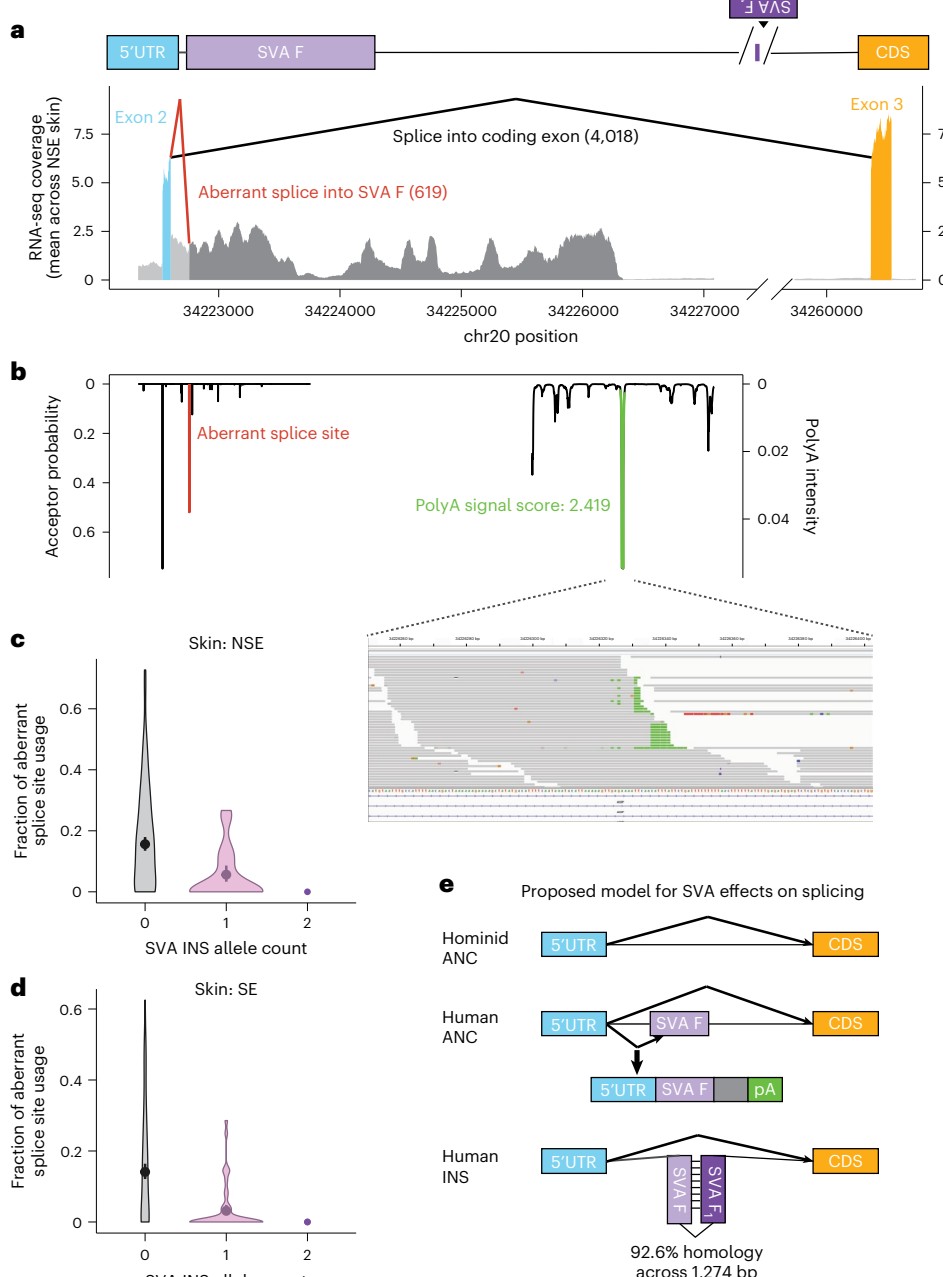

**Fig. 4 | Aberrant splicing and early polyadenylation of *ASIP* transcripts from haplotypes without SVA F$_1$ insertion. a**, RNA-seq coverage depth at *ASIP* (averaged across NSE skin samples), annotated with splice junctions from the exon 2 (5′ UTR) splice donor. Most split reads support the canonical splice junction to exon 3 ($n$ = 4,018 reads; black junction), but a substantial minority support an aberrant splice junction into an acceptor site in a nearby (nonpolymorphic) SVA F element ($n$ = 619 reads; red junction). **b**, Computationally predicted splice acceptors (SpliceAI[36]) and polyadenylation signals (APARENT[37]). Inset, RNA-seq read alignments containing soft-clipped poly(A) sequences at the predicted polyadenylation peak. **c**, Fraction of splice junctions from exon 2 that aberrantly splice into the acceptor site in the nearby SVA F element (versus splicing to exon 3), stratified by SVA F$_1$ insertion genotype in GTEx NSE skin samples. Only samples with greater than ten total reads supporting either splice junction are included in the violin plots ($n$ = 131)

to reduce noise from less informative samples. Centers, combined fraction of aberrant splicing across all samples with each SVA F$_1$ insertion genotype (total $n$ = 517); error bars, 95% CIs from bias-corrected and accelerated bootstrap. **d**, Analogous to **c**, for GTEx SE skin samples ($n$ = 158 for violin plots). Centers, combined fraction of aberrant splicing across all samples with each SVA F$_1$ insertion genotype (total $n$ = 605); error bars, 95% CIs from bias-corrected and accelerated bootstrap. **e**, Proposed model for the effects of the ancient SVA F insertion and the recent SVA F$_1$ insertion on splicing patterns of *ASIP* transcripts. The original hominid ancestral allele−lacking either retrotransposon−splices normally between alternate noncoding exon 2 and coding exon 3. Insertion of the SVA F element then causes a fraction of *ASIP* transcripts to splice aberrantly to the introduced acceptor site, leading to early polyadenylation. The subsequent SVA F$_1$ insertion then restores normal splicing to coding exon 3 in all transcripts, possibly by sequestering the splice acceptor or splice enhancer motifs.

European haplotypes at the *ASIP* locus, reaching up to 11% allele frequency in some European populations in a short period of time.

The timing and effects of these two retrotransposon insertions at *ASIP* seem broadly consistent with early and recent changes in human

pigmentation. The ancient SVA F insertion−which, despite being relatively young within the SVA F family (Methods), is present in Neanderthal genomes[40–43] (Supplementary Figs. 1 and 2)−probably contributed to increases in pigmentation early in human evolution, potentially

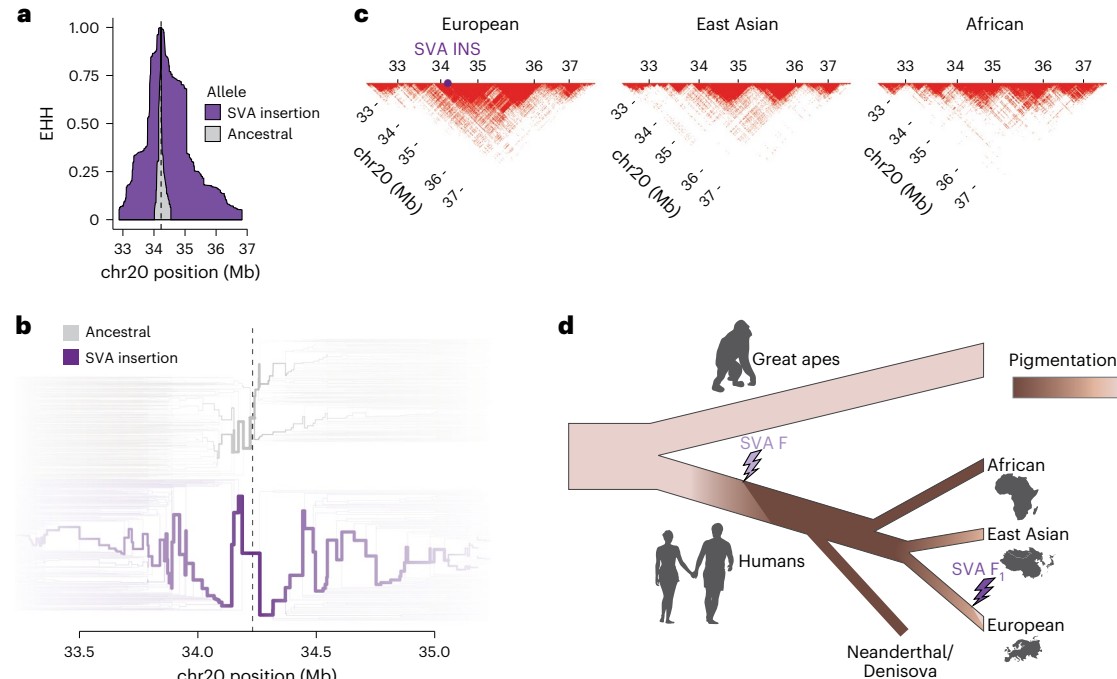

**Fig. 5 | Recent selection for the SVA $F_1$ insertion haplotype in ancestral European populations. a**, Extended haplotype homozygosity (EHH) plot for haplotypes with and without the SVA $F_1$ insertion in UKB ($n = 169,641$). The EHH value at a given variant is the probability that two haplotypes are homozygous at all variants between it and the focal variant[49] (here, the SVA $F_1$ insertion). Haplotypes with the SVA $F_1$ insertion have higher EHH in both directions, suggesting recent positive selection on the allele. **b**, Haplotype bifurcation diagram depicting haplotypes with and without the SVA $F_1$ insertion (1,000 haplotypes per group, selected randomly from the UKB analysis set). Bifurcations indicate SNPs that distinguish haplotypes, and line weights indicate proportions of haplotypes that carry each SNP allele. This diagram provides a haplotype-level representation of the comparatively reduced number of recombination events that have occurred on haplotypes containing the SVA $F_1$ insertion. **c**, LD in 5-Mb region surrounding *ASIP* (32.5–37.5 Mb; GRCh38 coordinates) in 1KGP superpopulations (excluding admixed African populations). For each superpopulation, 13,000 SNP/indel variants with MAF > 1% were sampled, and

the LD plot displays a red point for each pair of variants with $r^2 > 0.2$. Haplotypes sampled in populations with European genetic ancestry ($n = 1,006$) exhibit excess LD between variants in this region compared with African ($n = 1,002$) or East Asian ($n = 1,008$) haplotypes. The purple point indicates the relative position of the SVA $F_1$ insertion in European haplotypes. **d**, Evolution of hair and skin pigmentation in hominid lineages, with relative timing and pigmentation effects of each SVA insertion highlighted. Ancestral hominids and many extant great apes have light skin pigmentation, with UV protection conferred by denser body hair. Early human evolution involved increasing pigmentation and decreasing body hair. The ancient SVA F retrotransposon, which is shared by Neanderthals with modern humans on all continents, may have inserted during this period into the *ASIP* intron—decreasing *ASIP* expression and increasing pigmentation—and became fixed in *Homo sapiens*. Much more recently, a subsequent SVA $F_1$ insertion appeared and expanded in frequency (to several percent) within ancestral European populations, increasing *ASIP* expression and decreasing pigmentation.

helping to enable humans' concomitant loss of body hair. As modern humans later migrated around the world, pigmentation-lightening alleles appear to have emerged in several settings with reduced exposure to ultraviolet light[44,45]. The much more recent SVA $F_1$ insertion seems to have contributed to decreased pigmentation in a subset of individuals within ancestral European populations, while also leading to an increase in sunburn frequency (OR = 1.34 [1.30–1.37]; $P = 1.4 \times 10^{-105}$) and skin cancer risk (Fig. 5d).

*ASIP* thus appears to provide an example of how a sequence of retrotransposon insertions at a single locus can modulate phenotype several times in a species' recent history. The fact that the effect of such a common, large polymorphism (3.3 kb) could remain unnoticed for 15 years (even after recent advances in retrotransposon association analysis[2]) speaks to the importance of fully integrating structural variants into genetic association analyses. Interestingly, on evolutionary timescales, expression of *ASIP* and its homologs seems to have been modulated primarily by structural mutations that caused pigmentation changes in diverse species[18–22]. It is interesting to consider the possibility that some loci could be prone, across tens of millions of years, to evolve via structural variation.

The causal mechanism we have proposed—in which the recent SVA $F_1$ insertion acts by mitigating a splicing hypofunction introduced by an ancient SVA F insertion—should be testable by future

experiments that insert the SVA $F_1$ element into cell types that express *ASIP*. Increasingly available single-cell RNA-seq data may help identify the responsible cell type, which previous work has suggested could be fibroblasts or melanocyte precursors in the dermis[11,46]. While a recent study found that ectopic expression of *ASIP* (due to a rare *ITCH–ASIP* gene fusion) yields a monogenic phenotype including obesity, overgrowth and light pigmentation[47], the common SVA insertion polymorphism had little-to-no association with anthropometric traits in UKB (Extended Data Fig. 9) despite ample power to find associations, and its effect on *ASIP* expression (in GTEx) was observed only in skin and tibial nerve tissues (Fig. 3 and Extended Data Figs. 4 and 5) (possibly reflecting shared descent of melanocytes and Schwann cells from a common neural crest-derived progenitor[48]). More broadly, these results highlight the potential for integrative analyses of newly available genetic data resources to yield new insights, even in loci that have been well studied.

## Online content

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

## Methods

### HGSVC2 genetic data

PacBio long-read assembly contigs and Bionano structural variant calls for individuals in HGSVC2 were downloaded from the 1KGP FTP site (ftp://ftp.1000genomes.ebi.ac.uk/vol1/ftp/data_collections/HGSVC2/release/v1.0/assemblies/ and ftp://ftp.1000genomes.ebi.ac.uk/vol1/ftp/data_collections/HGSVC2/working/20200219_Bionano_optical_map_SVs/). Assembled sequences around *ASIP* were extracted by building each set of haploid contigs (h1 and h2) into a BLAST database and using a region from *ASIP* lacking repetitive sequences (chr20:34224248–34225005 in GRCh38) as a search query for BLASTN (v.2.12.0). Dotplots were generated with FlexiDot (v.1.06)[50].

### 1KGP genetic data

High-coverage data for 2,504 unrelated 1KGP individuals[30] was sliced to obtain paired reads mapping to the genomic interval chr20:34221626–34232418. To genotype the SVA $F_1$ element present in the reference genome at chr20:34228123–34231419, we counted (1) discordant read pairs aligned within a window extending 1 kb in each direction from the SVA $F_1$ with template length (TLEN) exceeding 2.5 kb (indicative of the presence of the major allele not containing the SVA $F_1$); and (2) reads spanning the right breakpoint (chr20:34231418) (indicative of the minor allele with the SVA $F_1$). Individuals were genotyped as homozygous for having the SVA $F_1$ insertion (Hom-INS) if the number of reads overlapping the right breakpoint ($n_{INS}$) was greater than three times the number of reads with TLEN >2.5 kb ($n_{ANC}$), that is, $n_{INS} > 3n_{ANC}$. Individuals were genotyped as homozygous for not having the insertion (Hom-ANC) if the number of reads overlapping the right breakpoint was <0.25× the number of reads with TLEN >2.5 kb, that is, $n_{INS} < 0.25n_{ANC}$. The remaining individuals were genotyped as heterozygous for the insertion. This strategy was necessitated by low mappability throughout the SVA $F_1$ element, which precluded read-depth analysis in the region. Similarly, to search for individuals that could in theory lack the upstream SVA F element at chr20:34222626–34224238, we counted (1) discordant read pairs aligned within a window extending 1 kb in each direction from the SVA F with TLEN exceeding 1.25 kb (which would suggest the presence of an allele not containing the SVA F); and (2) reads spanning the right breakpoint (chr20:34224238) (indicative of the expected SVA F sequence).

Pairwise LD plots (Fig. 5c) were generated for each of the European, East Asian and African genetic ancestry superpopulations by first extracting variants in the region chr20:32.5–37.5 Mb (GRCh38 coordinates) with population minor allele frequency (MAF) >1% with bcftools (v.1.14)[48]. ASW and ACB populations were excluded from the African genetic ancestry set to avoid selecting variants that would have excessively long linkage due to recent admixture. For each superpopulation, 13,000 variants were then sampled randomly to yield even density in plotting, and pairwise $r^2$ matrices were computed with plink1.9 (v.1.90b6.26)[51].

### UKB genetic and phenotype data

Genotyping of the SVA $F_1$ insertion polymorphism was performed on read alignments at *ASIP* extracted from WGS data available for 199,956 UKB participants[32]. The same overall genotyping approach as above was used for UKB, with the slight modification that individuals were genotyped as Hom-INS if the read categories from before satisfied $n_{INS} > 10 + 3n_{ANC}$ and as Hom-ANC if $n_{INS} < 2 + 0.2n_{ANC}$. Different linear separators were used here based on observed differences in the relative presence of $n_{INS}$ and $n_{ANC}$ for each genotype, presumably due to slight differences in sequencing and alignment parameters (for example, average coverage, fragment length, bwa-mem options). In this much larger sample, a third genotype cluster with few discordant read pairs and many reads overlapping the right breakpoint (indicating homozygosity for the SVA $F_1$ insertion allele) became clearly defined. Because the UKB data set contains several hundred sibling pairs that share both

haplotypes IBD2 in this region (based on at most three mismatching SNP-array genotypes within a 2-Mb window centered at *ASIP*, computed using plink1.9 –genome), we could correlate genotype calls made between these sibling pairs to estimate genotyping accuracy.

From these 199,956 individuals, we first removed 11,953 who did not report White ethnicity. A further 18,027 individuals were then filtered to remove outliers (>6 s.d.) across the first ten genetic ancestry principal components (PCs) and to select one individual from each first- or second-degree related pair, as described[52]. An additional 335 individuals were then excluded who had WGS but were not present in the imp_v3 imputed genotype dataset[31]. This set of 169,641 individuals was then used for both genome-wide association analyses with BOLT-LMM (v.2.4.1)[53] as well as local association analyses with plink2 (v.2.00a3.7; Intel AVX2)[51] (see below). Phenotypes of self-reported pigmentation traits (tanning ability, skin color, hair color, childhood sunburn frequency) were obtained from UKB touchscreen questionnaire datafields, adjusting the coding of hair color to assign red hair a value of 1 and blonde hair a value of 2 to better follow the order corresponding to increasing eumelanin:pheomelanin ratio[54] and binarizing the sunburn phenotype to individuals with no instances of severe childhood sunburns and those with at least one instance. Phenotypes of anthropometric traits (height, body mass index (BMI), waist-hip ratio adjusted for BMI) were processed as previously described[55]. Phenotypes of skin cancer diagnoses (derived from cancer registry data, accessed 26 October 2022) were taken from the 10th revision of the International Statistical Classification of Diseases and Related Health Problems (ICD-10) codes C43 (melanoma), C44 (nonmelanoma) and the union of the two for all skin cancers (C43 + C44).

### UKB: local association analyses at *ASIP*

VCF files containing genotype calls from UKB WGS were processed by first splitting multiallelic variants into separate biallelic variants with bcftools. Association analyses were performed on variants with MAF > 0.001 using plink2 (v.2.00a3.7; Intel AVX2) using linear regression for quantitative traits (tanning ability, skin color, hair color, height, BMI and waist-hip ratio adjusted for BMI) and logistic regression for binary skin cancer traits (C43, C44 and C43 + C44) with standard covariates (age, age squared, sex, 20 genetic PCs and assessment center). LD with the SVA $F_1$ ($r^2$) was computed with plink1.9 (v.1.90b6.26).

### UKB: genome-wide association analyses

Genome-wide association analyses of skin color, tanning ability and any skin cancer (C43 + C44) were performed on imputed variants (imp_v3) for the same 169,641 individuals using linear regression with BOLT-LMM (v.2.4.1). The above covariates as well as genotyping array were included as covariates, and variants were filtered to MAF > 0.001 and INFO > 0.3. Manhattan plots were generated using variants with $P < 0.01$ using the qqman (v.0.1.8) package[56].

### UKB: fine-mapping of phenotype associations

The 5,000 top associating variants from the region chr20:32.5–37.5 Mb (GRCh38) were extracted from VCF files containing genotype calls from UKB WGS (again splitting multiallelic variants into separate biallelic variants with bcftools). Standard covariates (age, age squared, sex, 20 genetic PCs and assessment center) were regressed out from both phenotype and genotypes and provided as input to the susieR (v.0.12.35) package[33] for fine-mapping (allowing up to $L = 10$ causal variables).

### UKB: measuring extended haplotype homozygosity

SVA $F_1$ diploid genotypes were phased onto a prephased scaffold of SNP-array genotyped variants for the same 169,641 individuals using the phase_common tool from SHAPEIT5 (v.5.1.1)[57]. The SNP-array genotyped variants were then matched with variants present in the low-coverage 1KGP3 (ref. [58]) VCF, for which ancestral and derived alleles had been annotated previously. These phased haplotypes were

then used to measure extended haplotype homozygosity (EHH) and generate bifurcation plots comparing haplotypes with and without the SVA $F_1$ insertion with the rehh (v.3.2.2) package[59].

## 1KGP: estimation of SVA $F_1$ insertion age

Variants present in the low-coverage 1KGP3 VCF were first recoded with REF allele as the ancestral allele. SVA $F_1$ diploid genotypes were then phased onto these scaffolds using the phase_common tool from SHAPEIT5. These phased haplotypes were then used as input to Relate (v.1.2.1)[60] with 1KGP3 genomic mask to filter low mappability regions (20140520.chr20.pilot_mask.fasta.gz), modified to include the SVA $F_1$ position itself. The coalescence rates provided by Relate for each 1KGP subpopulation were used.

## Estimation of SVA F insertion age

We compared the sequence of the *ASIP* SVA F with the sequences of other SVA F elements in the GRCh38 reference, which could in theory allow estimation of insertion time based on the numbers of observed base pair differences within the Alu-like and SINE-R SVA sequence elements (chr20:34222672–34223023 and chr20:34223717–34224222 at *ASIP*) and the de novo mutation rate. We ultimately concluded that these sequence elements were insufficiently long to accumulate enough mutations to allow a precise date estimate. However, we discovered that the most closely related SVA F (chr2:169785008–169787441, matching 349 of 352 bp in the Alu-like region and 505 of 506 bp in the SINE-R region) is commonly polymorphic[61]. This suggests that, despite being fixed in modern humans and archaic hominins, the *ASIP* SVA F has probably been active relatively recently and may be a relatively younger SVA in the SVA F family (which is estimated to be ~3 million years old[62]).

## GTEx genetic and expression data

Genotyping of the SVA $F_1$ insertion polymorphism was performed as above on read alignments at *ASIP* from 838 donors with WGS available in the GTEx v.8 release. Because the separation of genotype clusters was less visually clear in GTEx WGS using the criteria above, the read groups informative of the two alleles were redefined more strictly as (1) discordant read pairs in which one read aligns before the SVA $F_1$ left breakpoint (POS < 34228123), the other aligns after the right breakpoint (PNEXT > 34231419), and the TLEN exceeds 2.5 kb (indicative of the presence of the major allele not containing the SVA $F_1$) and (2) reads aligning at least 5 bp on each side of the right breakpoint and lacking any soft-clipping (indicative of the minor allele with the SVA $F_1$). Using these criteria, samples with 0 reads with TLEN >2.5 kb were called as homozygous for the SVA $F_1$ insertion, and samples with fewer than three reads overlapping the right edge were called as homozygous for no SVA $F_1$ insertion.

## GTEx: eQTL associations

For gene expression association analyses, the *ASIP* transcripts per million (TPM) values for a given tissue were taken from the GTEx v.8 release (GTEx_Analysis_2017-06-05_v8_RNASeQCv1.1.9_gene_tpm.gct). As some of the alternate first exons of *ASIP* were not included in the GENCODE v.26 definitions used by GTEx for expression quantification, TPM values for each exon and intronic region were computed from RNA-seq read counts. Specifically, the read counts for each region were first determined by using RNA-seq reads filtered with samtools (v.1.15.1)[63] view for being in proper pair (-f 0x2), not failing platform/vendor quality checks (-F 0x200), having an alignment distance ≤6, and mapping quality of 255 (following GTEx; https://gtexportal.org/home/methods) as input for bedtools (v.2.27.1)[64] coverage with -split flag. Separately, the TPM sample-level normalization factor previously computed by GTEx and applied to all genes from a given biosample was derived from read counts (GTEx_Analysis_2017-06-05_v8_RNASeQCv1.1.9_gene_reads.gct) and TPM values

(GTEx_Analysis_2017-06-05_v8_RNASeQCv1.1.9_gene_tpm.gct) for *GAPDH* (computing its length as the sum of its exon lengths; gencode.v26.GRCh38.genes.gtf), as:

$$\text{TPM scaling factor} = \frac{\text{read counts}}{(\text{effective gene length})(\text{TPM})}$$

We used *GAPDH* to recover this biosample-level scaling factor since *GAPDH* is highly expressed across all tissues, but the choice of gene used here has a negligible effect on TPM scaling factor estimation. TPM values for each exon or intron region were then calculated by first normalizing read counts from above by the region's size before dividing by the derived sample scaling factor.

For both gene-level and exon/intron-level eQTL analyses, the TPM values were analyzed for association with WGS-derived biallelic SNP and indel variants (GTEx_Analysis_2017-06-05_v8_WholeGenomeSeq_838Indiv_Analysis_Freeze.SHAPEIT2_phased.MAF01.vcf.gz) as well as the SVA $F_1$ insertion. Analyses were performed using linear models including all GTEx v8 covariates, and conditional analyses that additionally included SVA $F_1$ insertion genotype as a covariate were also performed.

Allelic fold change (aFC) was estimated as described[65]. First, in the linear regression

$$y = \beta_0 + \beta_g g + \boldsymbol{\beta}_{cov} \mathbf{X}_{cov} + \varepsilon$$

the intercept, $\beta_0$, estimates the expression level of two reference alleles, and the genotype effect size, $\beta_g$, estimates the difference of expression between alleles (alternate − reference), where $g$ are genotypes across donors, $\mathbf{X}_{cov}$ are covariates across donors, $\boldsymbol{\beta}_{cov}$ are the coefficients for each covariate, and $\varepsilon$ is noise. A point estimate of aFC can then be found as

$$\text{aFC} = \frac{2\beta_g}{\beta_0} + 1$$

where the estimated expression from reference and alternate alleles were each constrained to be positive. CIs were estimated with the adjusted bootstrap percentile (Bca) method with 10,000 replicates as implemented in R boot (v.1.3-28) library.

## GTEx: splicing QTL associations

Because the splicing phenotypes computed by GTEx consider a subset of splicing events at each locus[34], we quantified splice events across the region defined by the longest *ASIP* isoform. RNA-seq reads were first filtered with samtools as above, after which identification and quantification of splice junctions was performed with regtools (v.0.5.2)[66] junctions extract -a 8 -m 50 -M 500000 -s XS. This was first run on a merged bam from all GTEx biosamples corresponding to a given tissue to identify a set of nonspurious junctions ($n > 10$ observations total). The same regtools command was then run on each individual bam file, and the abundance of each junction seen in the merged set was recorded as individual-level quantification.

The fraction of reads aberrantly splicing into the SVA F splice acceptor was measured as the number of split reads supporting the junction between exon 2 and the splice acceptor within the SVA F divided by that plus the number of split reads supporting the junction between exons 2 and 3. For sQTL association analyses, the log-fraction of observed splice junctions for samples with at least one read spliced from exon 2 (adding a pseudocount of 1 to each junction count) was analyzed for association using the same approach as the eQTL analyses.

## Splice acceptor prediction

SpliceAI[36] (v.1.3.1) was run on the GRCh38 sequence in the region of the *ASIP* intron centered roughly on the nonpolymorphic SVA F element,

chr20:34221336–34224758, such that plotted SpliceAI predictions (Fig. 4b) considered at least 1 kb of sequence context on each side.

## Polyadenylation site prediction

APARENT[37] (v.0.1) was run using the aparent_large_lessdropout_all_libs_no_sampleweights model on a region of the *ASIP* intron centered roughly on the fall-off in RNA-seq read coverage between the non-polymorphic SVA F element and the SVA $F_1$ insertion, chr20:34224584–34228084, such that plotted predictions (Fig. 4b) considered at least 1 kb of sequence context on each side.

## Cloning of CAG SVA splicing construct

pCAGEN[39] was a gift from Connie Cepko (Addgene plasmid no. 11160; http://n2t.net/addgene:11160; RRID:Addgene_11160). mGreenLantern was synthesized as a gBlock from Integrated DNA Technologies. This was cloned into pCAGEN at the *Eco*RI and *Not*I cut sites using standard methods to yield pCAG-mGL. The human fixed SVA F was amplified from genomic DNA from 1KGP individual NA12878 using primers SVA_F and SVA_R. This amplicon then had pCAGEN-derived sequences added for Gibson assembly (New England Biolabs, cat. no. E2621S) by nested PCR with primers SVA_CAG_F and SVA_CAG_R to yield pCAG-mGL_SVA. All primer sequences can be found in Supplementary Table 1.

## Expression of CAG SVA construct, identification of splice site and reverse transcription digital droplet PCR

Plasmid pCAG-mGL_SVA was transfected into HEK293T cells (Takara Bio, cat. no. 632180) with Lipofectamine 3000 (Thermo Fisher Scientific, cat. no. L3000001) in six wells of two separate plates for a total of 12 replicates. Cells were given 24 h to express the construct before RNA was collected using Qiagen RNeasy columns (Qiagen, cat. no. 74104). To determine the exact location of the introduced splice junction, RNA was first converted to cDNA with oligo dT primers before amplifying the region spanning the expected junction between chicken beta-actin exon and SVA with CAG_bactin_fwd and CAG_SVA_rev primers. The amplicon matched the expected size (88 bp) and was Sanger sequenced with the CAG_SVA_rev primer.

To measure the relative amount of splicing into the inserted SVA F sequence, two ddPCR assays were designed: The first assay measures normal splicing between chicken beta-actin and rabbit beta-globin exons using HEX-labeled probe CAG_mGL_HEX with primers CAG_bactin_fwd and CAG_mGL_rev. The second assay measures splicing from chicken beta-actin exon to acceptor in SVA F using FAM-labeled probe CAG_SVA_FAM with primers CAG_bactin_fwd and CAG_SVA_rev. RNA was used as input with Bio-Rad One-Step RT-ddPCR Advanced Kit for Probes (Bio-Rad, cat. no. 1864022), with the optimal concentration of RNA input first identified by dilution series. Estimated Poisson-corrected concentrations of splicing into the SVA F were normalized by the sum of concentrations seen for both assays to yield an estimate of fraction spliced into the SVA F using QuantaSoft software (v.1.7). All primer and probe sequences can be found in Supplementary Table 1.

## Reporting summary

Further information on research design is available in the Nature Portfolio Reporting Summary linked to this article.

## Data availability

The following data resources are available by application: UKB (http://www.ukbiobank.ac.uk/) and GTEx (https://gtexportal.org/ under dbGaP accession no. phs000424.v9.p2). The following data resources are publicly available: 1KGP 30× coverage (https://www.internationalgenome.org/data-portal/data-collection/30x-grch38) and HGSVC2 (https://www.internationalgenome.org/data-portal/data-collection/hgsvc2).

## Code availability

The following publicly available software resources were used: BLAST (v.2.12.0, https://ftp.ncbi.nlm.nih.gov/blast/executables/blast+/2.12.0/), FlexiDot (v.1.06, https://github.com/molbio-dresden/flexidot), bcftools (v.1.14, http://www.htslib.org/), samtools (v.1.15.1, http://www.htslib.org/), plink (v.1.90b6.26 and v.2.00a3.7, https://www.cog-genomics.org/plink/), BOLT-LMM (v.2.4.1, https://alkes-group.broadinstitute.org/BOLT-LMM/), susieR (v.0.12.35, https://stephenslab.github.io/susieR/), qqman (v.0.1.8), https://cran.r-project.org/web/packages/qqman/index.html), SHAPEIT5 (v.5.1.1, https://odelaneau.github.io/shapeit5/), rehh (v.3.2.2, https://cran.r-project.org/web/packages/rehh/index.html), regtools (v.0.5.2, https://regtools.readthedocs.io/en/latest/), bedtools (v.2.27.1, https://bedtools.readthedocs.io/en/latest/), SpliceAI (v.1.3.1, https://github.com/Illumina/SpliceAI), APARENT (v.0.1, https://apa.cs.washington.edu/) and Relate (v.1.2.1, https://myersgroup.github.io/relate/index.html). The following proprietary software resources were used: QuantaSoft (v.1.7, https://www.bio-rad.com/en-us/life-science/digital-pcr/qx200-droplet-digital-pcr-system/quantasoft-software-regulatory-edition). Custom code used to generate results in this study is available via Zenodo at https://doi.org/10.5281/zenodo.10407629 (ref. 67).

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

## Acknowledgements

We thank A. Akbari, A. Barton, G. Genovese and M. Florio for helpful discussions; C. Usher for edits to text and figures and A. Arguello, A. Lewis and S. Hyman for helpful comments on the manuscript. This research was conducted using the UKB Resource under application number 40709. N.K. was supported by a US National Institutes of Health (NIH) training grant T32 HG002295. M.L.A.H. was supported by US NIH Fellowship F32 HL160061. R.E.M. was supported by US NIH grant K25 HL150334. S.A.M. was supported by US NIH grant R01 HG006855. P.-R.L. was supported by US NIH grants DP2 ES030554, R56 HG012698 and R01 HG013110 and a Burroughs Wellcome Fund Career Award at the Scientific Interfaces. The funders had no role in study design, data collection and analysis, the decision to publish or the preparation of the manuscript. The content is solely the responsibility of the authors and does not necessarily represent the official views of the NIH. Computational analyses were performed on the O2 High Performance Compute Cluster supported by the Research Computing Group at Harvard Medical School (http://rc.hms.harvard.edu) and on the UKB Research Analysis Platform.

## Author contributions

N.K., S.A.M. and P.-R.L. conceived the study design. N.K., M.L.A.H., R.E.M. and P.-R.L. did the computational analyses. N.K. and E.G. did the in vitro experiments. N.K., S.A.M. and P.-R.L. wrote the manuscript with contributions from all authors.

## Competing interests

The authors declare no competing interests.

## Additional information

**Extended data** is available for this paper at https://doi.org/10.1038/s41588-024-01841-4.

**Correspondence and requests for materials** should be addressed to Nolan Kamitaki, Steven A. McCarroll or Po-Ru Loh.

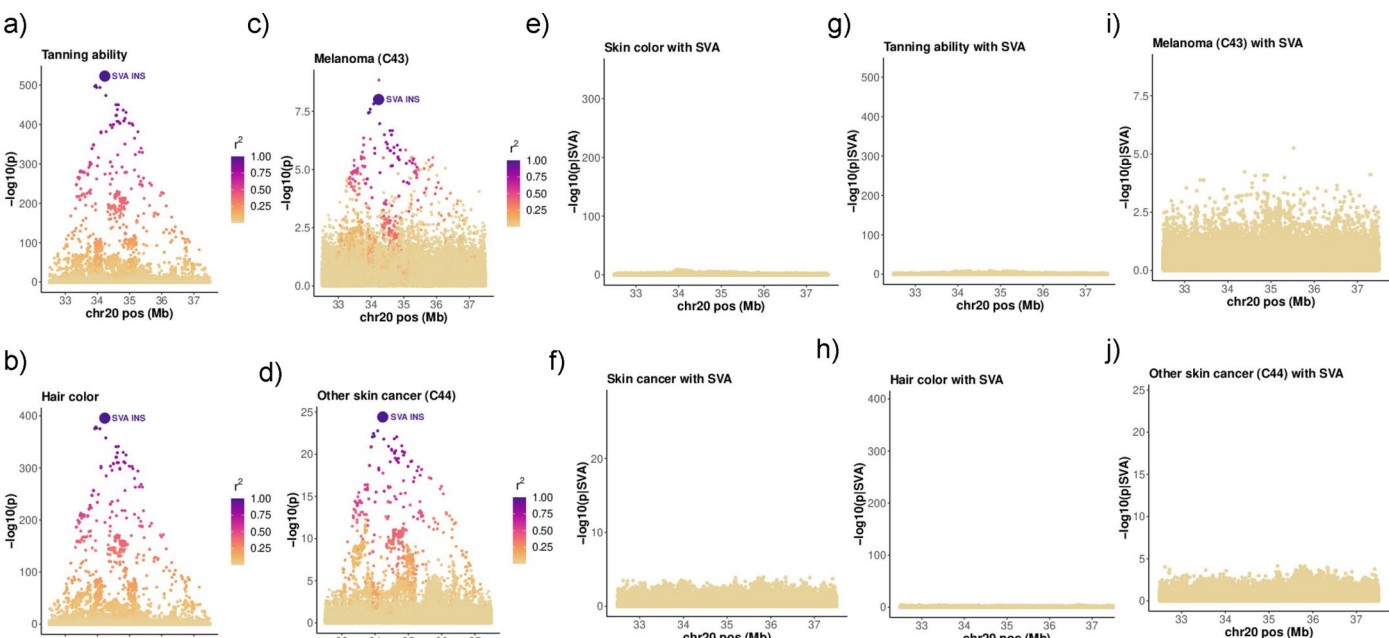

**Extended Data Fig. 1 | Associations of SVA F$_1$ insertion and nearby variants to pigmentation phenotypes in UK Biobank. a-d,** Local association plots in a 5-Mb window surrounding *ASIP* for (**a**) self-reported tanning ability ($n = 166,404$), (**b**) self-reported hair color ($n = 167,310$), (**c**) melanoma (C43 ICD-10 code, $n = 169,635$), and (**d**) other skin cancers including basal and squamous cell carcinomas (C44 ICD-10 code, $n = 169,635$). Association strengths track with linkage disequilibrium with the SVA F$_1$ insertion (yellow-to-purple shading), indicated by the large purple dot. **e-j,** Conditional association plots for SNPs and indels after including SVA F$_1$ genotype as a covariate for (**e**) self-reported skin color ($n = 167,568$), (**f**) any skin cancer (C43 or C44 ICD-10 codes, $n = 169,635$), (**g**) self-reported tanning ability ($n = 166,404$), (**h**) self-reported hair color ($n = 167,310$), (**i**) melanoma (C43 ICD-10 code, $n = 169,635$), and (**j**) other skin cancers including basal and squamous cell carcinomas (C44 ICD-10 code, n = 169,635).

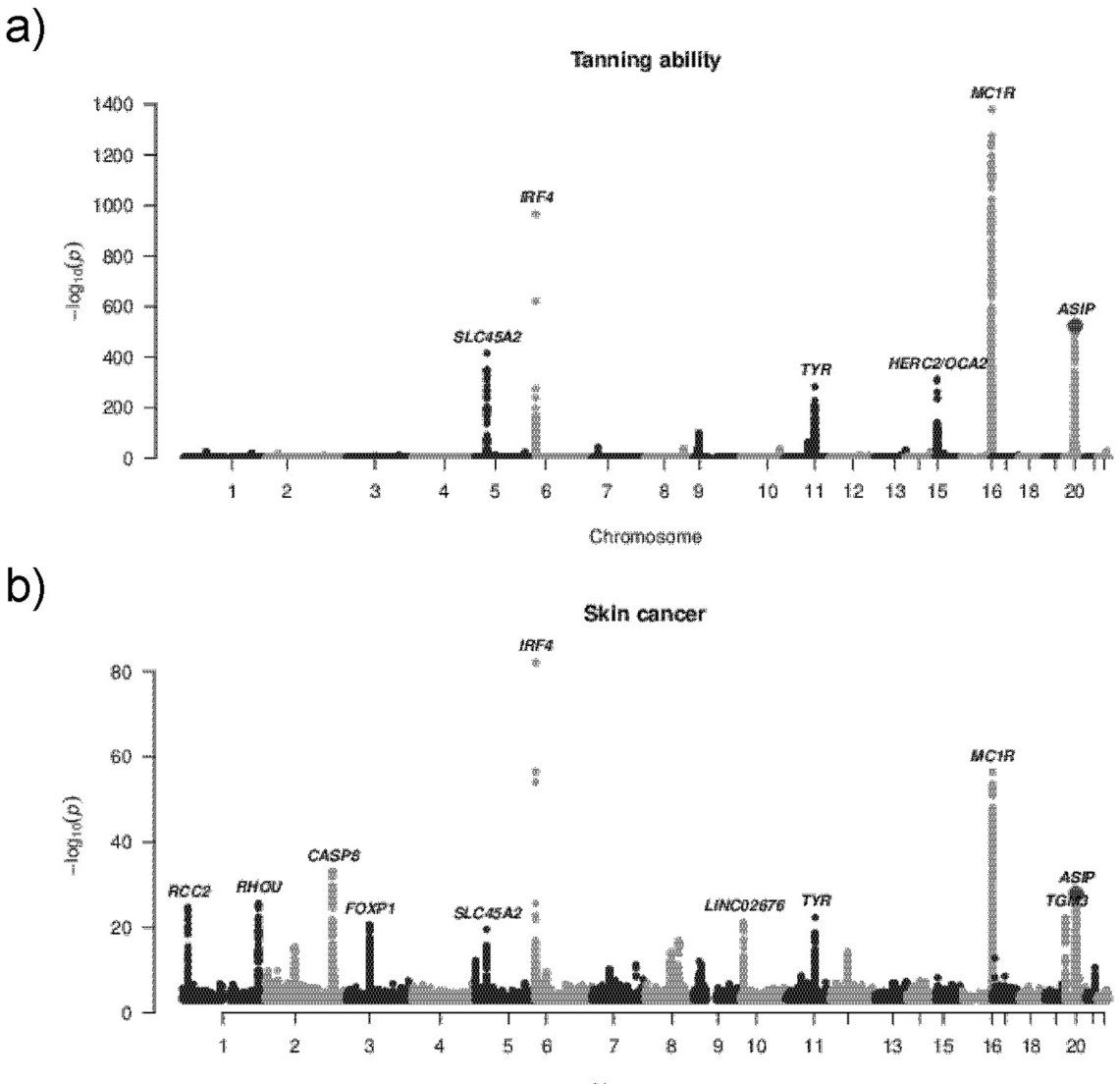

**Extended Data Fig. 2 | Genome-wide associations with tanning ability and skin cancer risk in UK Biobank. a,** Associations from linear regression across all imputed variants with self-reported tanning ability ($n$ = 166,404). **b,** Associations from linear regression across all imputed variants with any type of skin cancer (union of C43 and C44 ICD-10 codes, $n$ = 169,635).

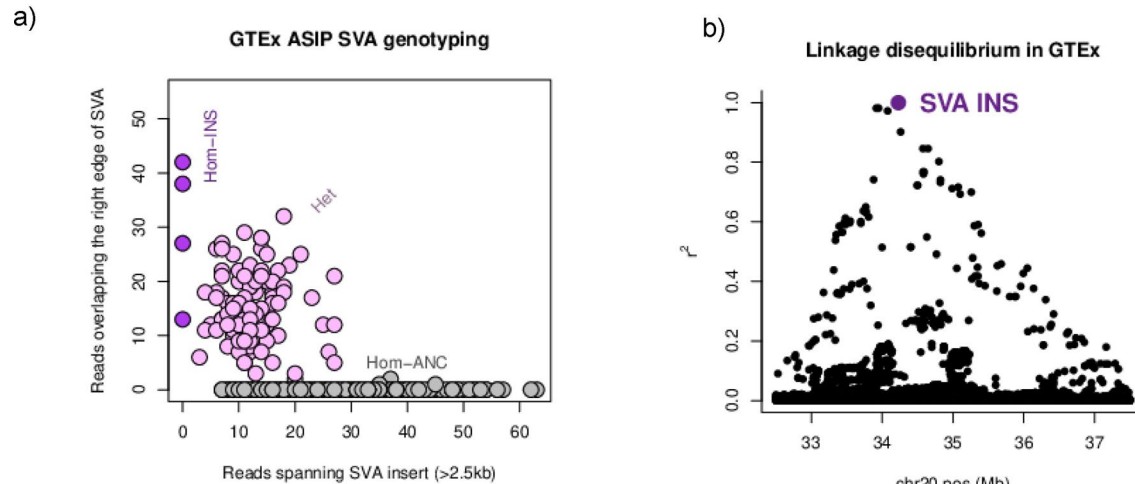

**Extended Data Fig. 3 | Genotyping of SVA F$_1$ insertion in GTEx cohort. a**, Genotyping of 838 GTEx donors with whole-genome sequencing. **b**, Linkage disequilibrium ($r^2$) of the SVA F$_1$ insertion with variants between 32.5 Mb to 37.5 Mb on chromosome 20 across GTEx donors.

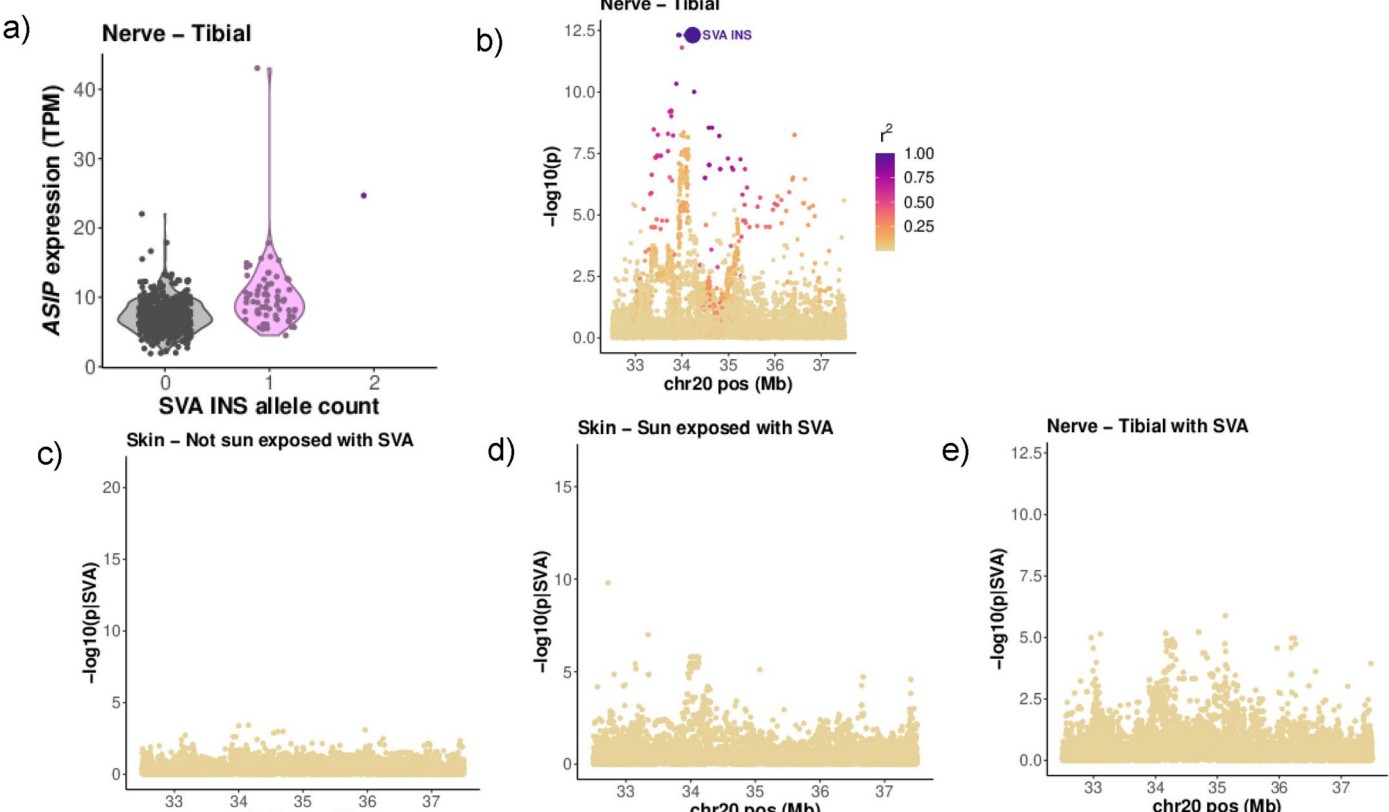

**Extended Data Fig. 4 | Associations of SVA $F_1$ insertion and nearby variants to expression of *ASIP* in skin and tibial nerve. a**, *ASIP* gene expression in GTEx tibial nerve samples ($n = 532$), stratified by SVA $F_1$ insertion genotype. Tibial nerve was the only other tissue that appeared to have evidence of the same eQTL. TPM, transcripts per million. **b**, Local association plot for *ASIP* gene expression in tibial nerve samples ($n = 532$) in the region 32.5 Mb to 37.5 Mb on chromosome 20. Association strengths track with linkage disequilibrium with the SVA $F_1$ insertion (yellow-to-purple shading), indicated by the large purple dot. **c**, Conditional association plot for *ASIP* gene expression in GTEx skin (not sun-exposed, NSE) samples ($n = 517$) after including SVA $F_1$ genotype as a covariate. **d**, As in **c**, but for skin (sun-exposed, SE) samples ($n = 605$). **e**, As in **c**, but for tibial nerve samples ($n = 532$).

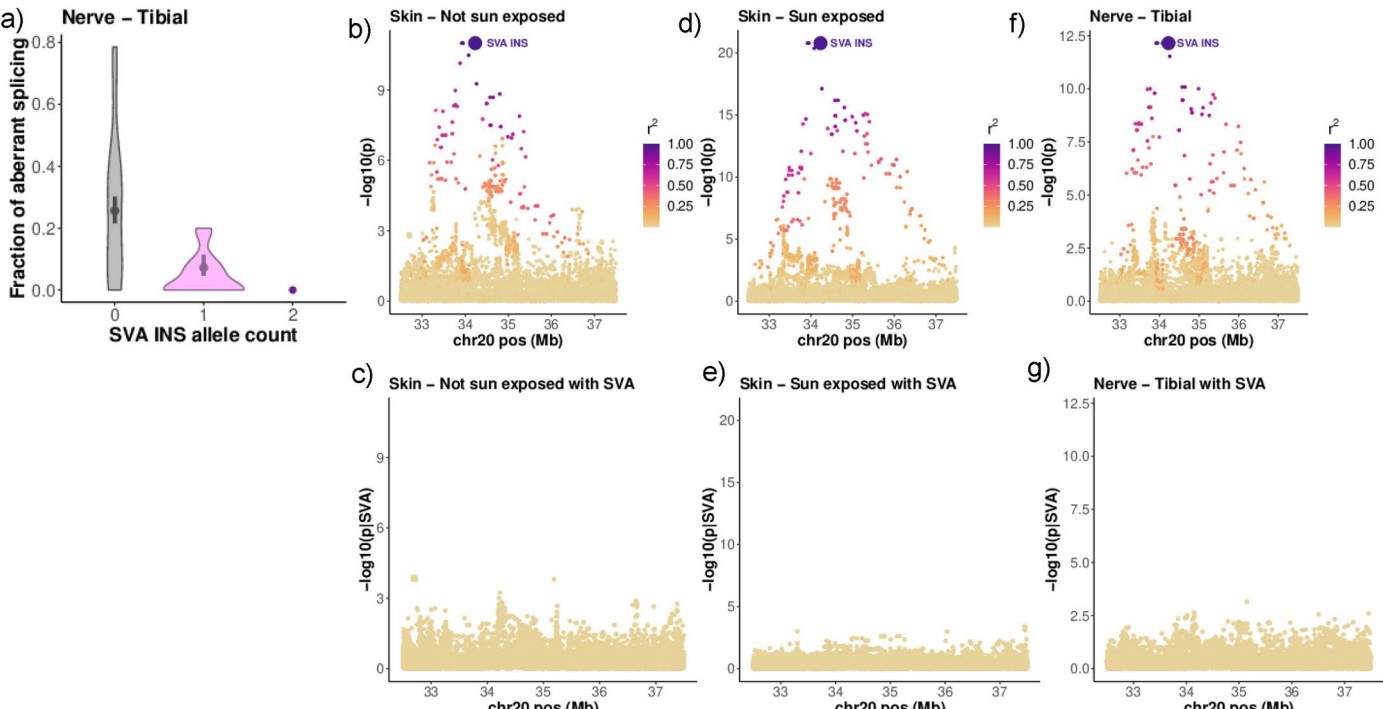

**Extended Data Fig. 5 | Associations of SVA $F_1$ insertion and nearby variants to aberrant *ASIP* splice junction usage in skin and tibial nerve. a**, Fraction of splice junctions from exon 2 that aberrantly splice into the acceptor site in the nearby SVA F element (versus splicing to exon 3), stratified by SVA $F_1$ insertion genotype in GTEx tibial nerve samples. Only samples with greater than 10 total reads supporting either splice junction are included in the violin plot ($n = 33$) to reduce noise from less informative points. Centers: combined fraction of aberrant splicing across all samples with each SVA $F_1$ insertion genotype (total $n = 532$); error bars: 95% CIs from bias-corrected and accelerated bootstrap.

**b**, Local association plot for aberrant splice junction usage in skin (not sun-exposed, NSE) samples ($n = 433$) in the region 32.5 Mb to 37.5 Mb on chromosome 20. Association strengths track with linkage disequilibrium with the SVA $F_1$ insertion (yellow-to-purple shading), indicated by the large purple dot. **c**, As in **b**, but after including SVA $F_1$ genotype as a covariate. **d**, As in **b**, but for skin (sun-exposed, SE) samples ($n = 497$). **e**, As in **d**, but after including SVA $F_1$ genotype as a covariate. **f**, As in **b**, but for tibial nerve samples ($n = 364$). **g**, As in **f**, but after including SVA $F_1$ genotype as a covariate.

a)

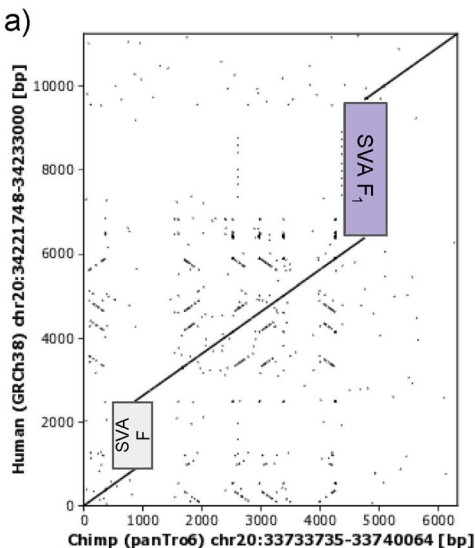

b)

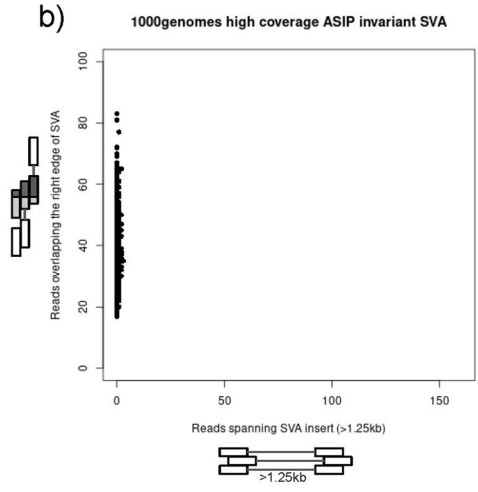

**Extended Data Fig. 6 | Characterization of non-polymorphic SVA F element upstream of SVA F₁ insertion. a**, Pairwise sequence alignment dot plot of GRCh38 human reference vs. panTro6 chimpanzee reference at *ASIP*. The human reference contains an SVA F element upstream of the polymorphic SVA F₁ insertion, neither of which are present in chimpanzee. **b**, Assessment of SVA F presence in 1KGP individuals. Similar to the genotyping approach we used for the polymorphic SVA F₁ insertion, we counted reads overlapping the right edge of the SVA F element (indicating presence of at least one allele containing the SVA F) and discordant reads with a fragment size approaching the length of the SVA F element (1.6 kb) (which would indicate the presence of an allele lacking the SVA F). All individuals in 1KGP appear to carry the SVA F on both *ASIP* alleles.

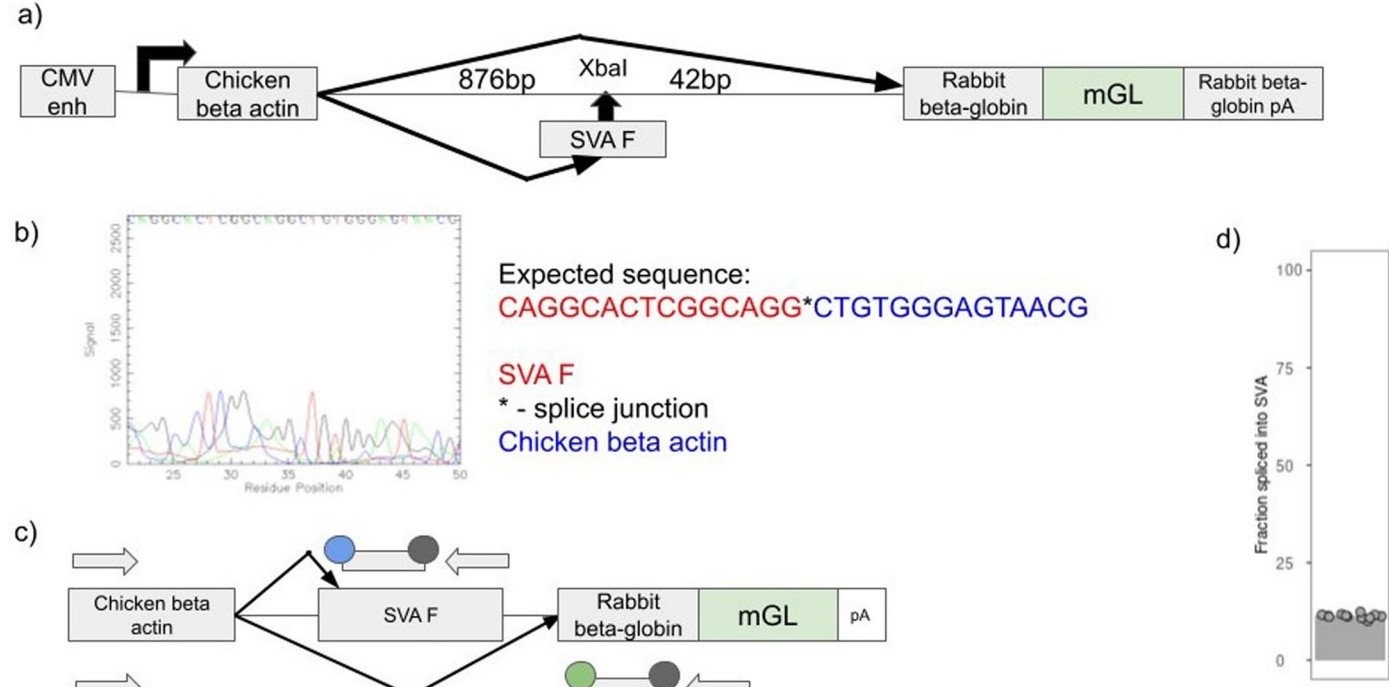

**Extended Data Fig. 7 | Construct to measure splicing into upstream SVA F element *in vitro*. a**, Design of base construct, pCAG-mGL, and relative position of introduced SVA F sequence in the hybrid intron of the CAG promoter at XbaI restriction site. **b**, Sanger sequencing results from transcripts produced by pCAG-mGL_SVA construct and match to the expected sequence that would arise from splicing from the upstream chicken beta-actin exon (in blue) to the aberrant splice acceptor within the SVA F element (in red) observed in GTEx RNA-seq at *ASIP*. Note that the sequence is antisense to the transcript. **c**, Design of RT-ddPCR assays to measure relative splicing from the chicken beta-actin exon to introduced splice acceptor in SVA F versus downstream rabbit beta-globin exon. The arrows represent forward and reverse primers and the rectangles with circles represent quenched fluorescently labeled probes, where the blue circle is a FAM fluorophore, the green circle is a HEX fluorophore, and the gray circles are quenchers that are cleaved during polymerase extension. **d**, Fraction of splicing into the introduced SVA F element in pCAG-mGL_SVA construct (*n* = 12 replicates). Each point indicates the measured value in a replicate, with the bar indicating the mean fraction.

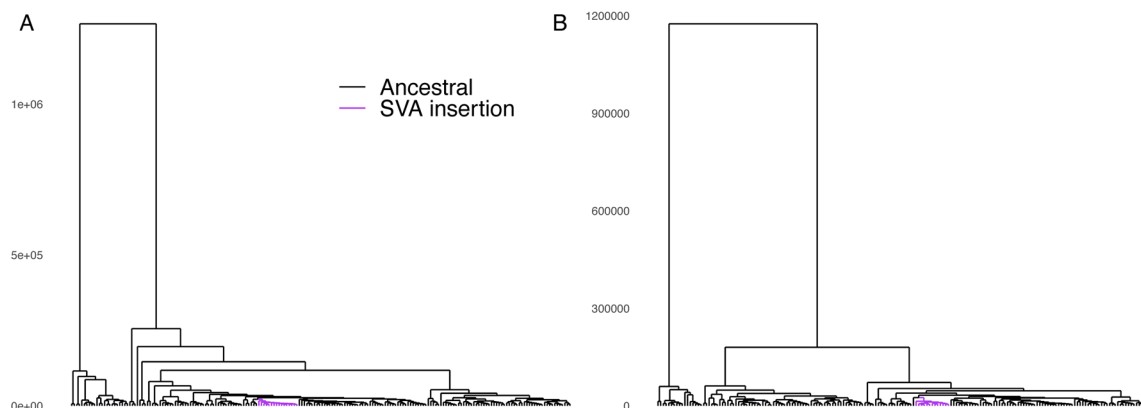

**Extended Data Fig. 8 | Genealogy of haplotypes at the *ASIP* SVA F$_1$ insertion. a,b**, Coalescent trees estimated by Relate for (**a**) CEU ($n = 202$) and (**b**) GBR ($n = 186$) haplotypes at the SVA F$_1$ insertion site. The purple branch contains all haplotypes carrying the SVA F$_1$ insertion. Age (in years) on the *y*-axis assumes a generation time of 28 years.

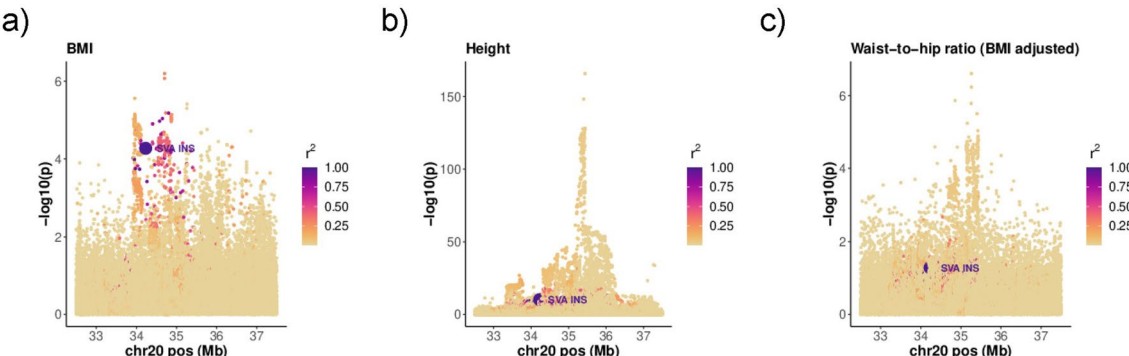

**Extended Data Fig. 9 | Associations of SVA F$_1$ insertion and nearby variants to anthropometric phenotypes in UK Biobank. a-c**, Local association plots in a 5-Mb window surrounding *ASIP* for (**a**) BMI (*n* = 169,052), (**b**) height (*n* = 169,239), and (**c**) waist-hip ratio adjusted for BMI (*n* = 169,285). Only the associations with height reach genome-wide significance, but the association pattern does not appear to colocalize with linkage disequilibrium with the SVA F$_1$ insertion (yellow-to-purple shading).

# Reporting Summary

## Statistics

For all statistical analyses, confirm that the following items are present in the figure legend, table legend, main text, or Methods section.

| n/a | Confirmed | |
|---|---|---|
| ☐ | ☒ | The exact sample size (*n*) for each experimental group/condition, given as a discrete number and unit of measurement |
| ☒ | ☐ | A statement on whether measurements were taken from distinct samples or whether the same sample was measured repeatedly |
| ☐ | ☒ | The statistical test(s) used AND whether they are one- or two-sided <br> *Only common tests should be described solely by name; describe more complex techniques in the Methods section.* |
| ☐ | ☒ | A description of all covariates tested |
| ☒ | ☐ | A description of any assumptions or corrections, such as tests of normality and adjustment for multiple comparisons |
| ☐ | ☒ | A full description of the statistical parameters including central tendency (e.g. means) or other basic estimates (e.g. regression coefficient) AND variation (e.g. standard deviation) or associated estimates of uncertainty (e.g. confidence intervals) |
| ☐ | ☒ | For null hypothesis testing, the test statistic (e.g. *F*, *t*, *r*) with confidence intervals, effect sizes, degrees of freedom and *P* value noted <br> *Give P values as exact values whenever suitable.* |
| ☒ | ☐ | For Bayesian analysis, information on the choice of priors and Markov chain Monte Carlo settings |
| ☒ | ☐ | For hierarchical and complex designs, identification of the appropriate level for tests and full reporting of outcomes |
| ☐ | ☒ | Estimates of effect sizes (e.g. Cohen's *d*, Pearson's *r*), indicating how they were calculated |

*Our web collection on statistics for biologists contains articles on many of the points above.*

## Software and code

Policy information about availability of computer code

Data collection    QuantaSoft (v1.7) as associated with QX200 droplet reader was used to collect fluorescence measurements for digital droplet PCR.

Data analysis       The following publicly available software resources were used: BLAST (v2.12.0, https://ftp.ncbi.nlm.nih.gov/blast/executables/blast+/2.12.0/), FlexiDot (v1.06, https://github.com/molbio-dresden/flexidot), bcftools (v1.14, http://www.htslib.org/), samtools (v1.15.1, http://www.htslib.org/), plink (v1.90b6.26 and v2.00a3.7, https://www.cog-genomics.org/plink/), BOLT-LMM (v2.4.1, https://alkesgroup.broadinstitute.org/BOLT-LMM/), susieR (v0.12.35, https://stephenslab.github.io/susieR/), qqman (v0.1.8), https://cran.r-project.org/web/packages/qqman/index.html), SHAPEIT5 (v5.1.1, https://odelaneau.github.io/shapeit5/), rehh (v3.2.2, https://cran.r-project.org/web/packages/rehh/index.html), regtools (v0.5.2, https://regtools.readthedocs.io/en/latest/), bedtools (v2.27.1, https://bedtools.readthedocs.io/en/latest/), SpliceAI (v1.3.1, https://github.com/Illumina/SpliceAI), APARENT (v0.1, https://apa.cs.washington.edu/), and Relate (v1.2.1, https://myersgroup.github.io/relate/index.html). Custom code used to generate results in this study has been deposited in Zenodo doi:10.5281/zenodo.10407629.

For manuscripts utilizing custom algorithms or software that are central to the research but not yet described in published literature, software must be made available to editors and reviewers. We strongly encourage code deposition in a community repository (e.g. GitHub). See the Nature Portfolio guidelines for submitting code & software for further information.

## Data

Policy information about availability of data

All manuscripts must include a data availability statement. This statement should provide the following information, where applicable:

- Accession codes, unique identifiers, or web links for publicly available datasets
- A description of any restrictions on data availability
- For clinical datasets or third party data, please ensure that the statement adheres to our policy

The following data resources are available by application: UK Biobank (http://www.ukbiobank.ac.uk/) and Genotype-Tissue Expression (GTEx) project (https://gtexportal.org/, under dbGaP accession number phs000424.v9.p2). The following data resources are publicly available: 1000 Genomes Project (1KGP) 30x coverage (https://www.internationalgenome.org/data-portal/data-collection/30x-grch38) and Human Genome Structural Variation Consortium, Phase 2 (HGSVC2) (https://www.internationalgenome.org/data-portal/data-collection/hgsvc2).

## Research involving human participants, their data, or biological material

Policy information about studies with human participants or human data. See also policy information about sex, gender (identity/presentation), and sexual orientation and race, ethnicity and racism.

| | |
|---|---|
| Reporting on sex and gender | Sex was used as a covariate in several analyses, but no values directly pertaining to sex are reported. |
| Reporting on race, ethnicity, or other socially relevant groupings | A subset of individuals within the UK Biobank cohort that self-identified as "white" and were not outliers (>6 standard deviations) on the first 10 genetic ancestry principal components were used for phenotype associations. Populations within 1000 genomes were analyzed according to their previously published ancestry groupings. GTEx contains individuals of multiple reported races, but this information was not used in this study. |
| Population characteristics | For phenotype associations, age (and age squared), sex, the top 20 genetic ancestry principal components, and intake assessment center were used as covariates.<br><br>For expression and splicing analyses, all GTEx v8 covariates were used - 5 genetic ancestry principal components, sex, PCR or PCR-free WGS preparation, Illumina sequencing platform (HiSeq 2000 or HiSeq X), and any inferred PEER covariates specific for each tissue and analysis type (expression or splicing). |
| Recruitment | Individuals and biosamples were not obtained for this study and their recruitment is as described in prior publications (cited in current work). |
| Ethics oversight | Individuals and biosamples were not obtained for this study and local IRBs at each institution approved the collections and patient-consent materials, as described in the earlier papers on these cohorts (cited in current work). Datasets were used as approved for research plans as stated in applications to each: UK Biobank Resource application #40709 and project #28875 to dbGaP accession phs000424.v9.p2 (GTEx) |

Note that full information on the approval of the study protocol must also be provided in the manuscript.

# Field-specific reporting

Please select the one below that is the best fit for your research. If you are not sure, read the appropriate sections before making your selection.

☒ Life sciences    ☐ Behavioural & social sciences    ☐ Ecological, evolutionary & environmental sciences

For a reference copy of the document with all sections, see nature.com/documents/nr-reporting-summary-flat.pdf

# Life sciences study design

All studies must disclose on these points even when the disclosure is negative.

| | |
|---|---|
| Sample size | A set of 169,641 individuals in UK Biobank with WGS available were used for all analyses, with some samples having missing information for each phenotype as noted in the text. A set of 878 individuals in GTEx with WGS available were used for all analyses, with some samples having missing RNA-seq data for each tissue as noted in the text. A set of 1508 individuals in 1KGP were used to evaluate linkage disequilibrium in the region surrounding the ASIP locus in populations from three genetic ancestral backgrounds. A subset of individuals in 1KGP (n=194) were used to estimate allele genealogy. In all cases, no sample-size calculation was done to predetermine sample size and the maximum number of available samples were used. For phenotype associations (UK Biobank), we expected that the association would be sufficiently powered to allow for fine-mapping given 1) strength of association to linked variants, 2) linkage (r2=0.97), and 3) genotyping accuracy (r=0.997). For expression and splicing associations (GTEx), the number of samples was not expected to allow for the same level of fine-mapping against other genetic variants in high linkage, but were expected to be sufficient to confirm the same association pattern was observed and the lack of any secondary signal(s) that might be considered discordant with phenotype-genotype associations. For analysis of linkage disequilibrium, we expected that the number of samples would be sufficient to show ancestral population differences given the genomic range of SNPs previously seen to associate with pigmentation phenotypes in populations of European ancestry and lack of signal seen in similar association studies from other ancestries. For estimation of allele genealogy, we expected the sample size to be sufficient given the same samples were used to generate estimates for linked variants in the original publication. |

For in vitro experiments to confirm splicing into the SVA acceptor and measure in vitro splicing rate, no sample-size calculation was done to predetermine sample size as the main purpose was to observe any splice events into the SVA acceptor and confirm by Sanger sequencing. In terms of necessary sample size to accurately estimate the splicing rate, it would depend, in part, on unknown biological variability in splicing rate between SVA acceptor and downstream rabbit beta-globin exon acceptor, where the latter might cause variance to differ from that observed in endogenous ASIP exon acceptor use in skin tissue. We started with n=12 biological replicates which we determined to have a small enough sample variance that no additional replicates were needed for estimating rate.

**Data exclusions**

Established QC metrics were used to exclude some samples, genotypes, or sequencing data for analysis as described in previously published studies (cited in the current work). Samples from individuals in UK Biobank that requested to be withdrawn at the time of analysis were excluded. ASW and ACB populations within 1000 Genomes Project were excluded from the African genetic ancestry set in generating linkage blocks at the ASIP locus to avoid selecting variants that would have excessively long linkage due to recent admixture.

**Replication**

For phenotype-genotype associations, the multiple pigmentation phenotypes (skin color, tanning response, hair color, all skin cancer, non-melanoma skin cancer, and melanoma) each replicated both the overall pattern of association but also fine-mapping to the SVA as the strongest associating variant on the haplotype (with the exception of melanoma only, C43). Outside of these independently measured phenotypes from the UK Biobank cohort, these associations were not replicated or performed independently in another cohort.

Likewise, the two skin-derived tissues in GTEx (sun exposed lower leg and not sun exposed suprapubic) serve as replications for expression and splicing associations, as the measurements were from independent biosamples. The in vitro splicing assay had n=12 replicates, where all attempts were successful and included in Extended Data Figure 7.

**Randomization**

For UK Biobank, samples were collected in batches at different assessment centers at locations across the United Kingdom and these were encoded as indicator covariates in phenotype-genotype associations. For GTEx, samples were allocated into batches with different sample preparation methods (PCR or PCR-free) and sequencing machines (HiSeq 2000 or HiSeq X) due to changing methods during the course of their collection. These were likewise encoded as indicator covariates in expression- or splicing-genotype associations. No further randomization was done as all samples were used for each analysis.

**Blinding**

For all computational analyses, samples were listed with a randomized ID where association of measured genotype with trait (phenotype such as skin color or relative amount of transcript splicing) was only done at the point of final statistical analysis. Blinding was not done for the in vitro splicing assay where all samples were part of a single test group.

# Reporting for specific materials, systems and methods

We require information from authors about some types of materials, experimental systems and methods used in many studies. Here, indicate whether each material, system or method listed is relevant to your study. If you are not sure if a list item applies to your research, read the appropriate section before selecting a response.

## Materials & experimental systems

| n/a | Involved in the study |
|-----|----------------------|
| ☒ | Antibodies |
| ☐ | ☒ Eukaryotic cell lines |
| ☒ | Palaeontology and archaeology |
| ☒ | Animals and other organisms |
| ☒ | Clinical data |
| ☒ | Dual use research of concern |
| ☒ | Plants |

## Methods

| n/a | Involved in the study |
|-----|----------------------|
| ☒ | ChIP-seq |
| ☒ | Flow cytometry |
| ☒ | MRI-based neuroimaging |

## Eukaryotic cell lines

Policy information about cell lines and Sex and Gender in Research

**Cell line source(s)**

Lenti-X 293T (HEK293T clone) from Takara Bio USA; Lot#: AIY00015; Cat#: 632180

**Authentication**

Morphological match for type and in-house verification of SV40T antigen with genotyping PCR assay. No other standard authentication methods were performed (such as STR typing).

**Mycoplasma contamination**

Lack of mycoplasma contamination was done by Takara Bio USA as well as by members of receiving lab (McCarroll)

**Commonly misidentified lines**
(See ICLAC register)

None were used in this study, HEK293T is a derivative of HEK and has not been listed as commonly misidentified.

