## [Peer Review File · Nature Genetics]

Peer Review Information

Manuscript Title: A sequence of SVA retrotransposon insertions in *ASIP* shaped human pigmentation

Corresponding author name(s): Mr Nolan Kamitaki, Dr Po-Ru Loh, Dr Steven McCarroll

Reviewer Comments & Decisions:

Decision Letter, initial version:

25th September 2023

Dear Nolan,

Your Letter "A sequence of SVA retrotransposon insertions in *ASIP* shaped human pigmentation" has been seen by two referees. You will see from their comments below that, while they find your work of interest, they have raised several relevant points. We are interested in the possibility of publishing your study in Nature Genetics, but we would like to consider your response to these points in the form of a revised manuscript before we make a final decision on publication.

To guide the scope of the revisions, the editors discuss the referee reports in detail within the team, including with the chief editor, with a view to identifying key priorities that should be addressed in revision, and sometimes overruling referee requests that are deemed beyond the scope of the current study. In this case, we ask that you revise the presentation for clarity where needed, extend the analyses and functional studies where feasible as requested by the referees (e.g., estimate the age of the haplotype containing the SVA_F1 insertion, perform additional experiments to test the proposed model, etc.), and ensure the analysis code is made available through a public repository. We hope you will find this prioritized set of referee points to be useful when revising your study. Please do not hesitate to get in touch if you would like to discuss these issues further.

We therefore invite you to revise your manuscript taking into account all reviewer and editor comments. Please highlight all changes in the manuscript text file. At this stage, we will need you to upload a copy of the manuscript in MS Word .docx or similar editable format.

*2) If you have not done so already, please begin to revise your manuscript so that it conforms to our Letter format instructions, available here.
Refer also to any guidelines provided in this letter.

Please be aware of our guidelines on digital image standards.

[redacted]

We hope to receive your revised manuscript within 8-12 weeks. If you cannot send it within this time, please let us know.

Nature Genetics is committed to improving transparency in authorship. As part of our efforts in this direction, we are now requesting that all authors identified as 'corresponding author' on published papers create and link their Open Researcher and Contributor Identifier (ORCID) with their account on the Manuscript Tracking System (MTS), prior to acceptance. ORCID helps the scientific community achieve unambiguous attribution of all scholarly contributions. You can create and link your ORCID from the home page of the MTS by clicking on 'Modify my Springer Nature account'. For more information, please visit www.springernature.com/orcid.

Sincerely,
Kyle

Kyle Vogan, PhD

Senior Editor
Nature Genetics
<https://orcid.org/0000-0001-9565-9665>

Referee expertise:

Referee #1: Population genetics, mobile DNA elements, evolutionary biology

Referee #2: Population genetics, mobile DNA elements, evolutionary biology

Reviewers' Comments:

Reviewer #1:
Remarks to the Author:

In this study (NG-LE63191), the authors examined the functional impact of two SVA insertions on the ASIP gene and the resulting change in human skin pigmentation. The authors first identified a polymorphic SVA_F1 insertion in the intron 2 of the ASIP gene. The insertion is more prevalent in European populations than other populations and showed strong associations with pigmentation traits and skin cancer risk in UK Biobank samples. Using RNA-seq data from the GTEx project, the authors inferred that the SVA insertion increases ASIP expression by improving splicing of the functional isoform. The authors then identified an older, fixed SVA_F element in the same intron that introduced aberrant splicing of ASIP. Using an in vitro reporter assay, the authors showed that the SVA_F element introduce a splicing acceptor site competing with the canonical site. Lastly, the authors showed that the SVA_F1 insertion is on a young haplotype that appears to be quickly rise in frequency, suggesting the insertion is under positive selection.

Overall, this manuscript presents a well-organized study. The writing is clear, and the datasets and analyses were described sufficiently. The authors provide a potential explanation of the dynamic change in pigmentation during the human evolution, through the regulation of ASIP expression. I only have a few specific comments.

- Page 4: the SVA_F1 subfamily was defined in the article "5-Transducing SVA retrotransposon groups spread efficiently throughout the human genome" (PMID 19652014). Please cite the reference when first mentioning SVA_F1.
- Fig. 1f: It would be useful to label the index SNP rs6059655 in the plot.
- With the data described in the study, it is possible to estimate the age of the haplotype containing the SVA F1 insertion (see PMID 21816865 for an example). It can provide more insight into the time frame when the insertion occurred.

Reviewer #2:
Remarks to the Author:

This is a clearly written paper that describes fine mapping to identify a likely causal mutation that explains skin color variation and associated traits (i.e., skin cancer risk) among Europeans at the ASIP locus. Informatics analysis is given that argues that the causal mutation is a polymorphic SVA insertion, with a model where the polymorphic SVA insertion suppresses the creation of a nonfunctional transcript isoform through formation of an RNA hairpin with another SVA element that is present in all humans. This is consistent with prior knowledge on the potential of SVA insertions to cause splicing defects and leads to a model where an ancestral SVA disrupted ASIP function and that a second mutation found in some humans leads to a partial restoration. In other species, multiple types of mutations, including transposable elements, have been described at ASIP that play a role in coloration phenotypes.

This is a very nice effort to fine map a casual variant that would be further strengthened by additional functional data to test the model, including RNase protection assays to show that the proposed hairpin actually forms and more sophisticated minigene splicing assays to iteratively test effects of combinations of the two SVA elements. Such functional data would strengthen the paper, but I am not sure whether they would substantially enhance the broader significance of the finding.

The authors did an outstanding job in integrating genomics and biobank datasets to fine map the locus, but it is not clear to me that the lack of long read genomes and pangenome references are really the key resources for this type of study. For example, the SVA variant described here is relatively easy to discover and genotype since the most commonly used human reference genome has the 'insertion allele', making the variant appear as a "deletion" which does not require mobile-element specific informatics tools to analyze. In fact, this variant appears to have been first described in humans in 2007 (Korbel 2007 and Levy 2007, DGV accessions nsv435737 and essv4114058) and was found by every sequencing-based SV study since then that included Europeans, notably including the 1000 Genomes Project. Thus, it is a mystery why the statistical genetics community has failed to incorporate such SVs into their analysis; the authors here are the rare few in human genetics who are doing such.

The tradeoff of decreased pigmentation and increasing body hair is a nice idea but would benefit from evidence in terms of the timing of the relevant mutations.

Other minor comments to consider:

Line 106: if space allows further elaboration on the origin of the SVA subfamily may help, the mention of MAST1 exon is somewhat jarring to a casual reader without context.

Line 134: should be than all OTHER SNP and indel variants

Line 139: why is tanning response the best-powered?

Line 154+figures – any ideas for why Tibila Nerve also shows the expression association?

Line 157 Figure 3e colors do not seem to match – I do not see red for alternative 5' exons

Extended Data Figure 7c – the ddPCR design could benefit from more description to increase clarity for readers unfamiliar. What are the blue green and black circles, etc.?

Extended Data Figure 8. It is hard to see the IGV screen shots. They would benefit from higher resolution (no need to force 4 panels together in a supplementary figure) and by including the Repeat Masker track in the IGV view to visualize the location of SVA in the reference.

Methods: some inconsistency as to whether version numbers are listed for programs

Line 720/749. It is not clear why different criteria were employed in the two data sets.

Line 937 – scripts available upon request is frowned upon these days. It is not hard to put anything that is relevant in GitHub/Zenodo to share.

Author Rebuttal to Initial comments

Response to reviewers of NG-LE63191 (Kamitaki et al.)

Reviewer #1:

In this study (NG-LE63191), the authors examined the functional impact of two SVA insertions on the ASIP gene and the resulting change in human skin pigmentation. The authors first identified a polymorphic SVA_F1 insertion in the intron 2 of the ASIP gene. The insertion is more prevalent in European populations than other populations and showed strong associations with pigmentation traits and skin cancer risk in UK Biobank samples. Using RNA-seq data from the GTEx project, the authors inferred that the SVA insertion increases ASIP expression by improving splicing of the functional isoform. The authors then identified an older, fixed SVA_F element in the same intron that introduced aberrant splicing of ASIP. Using an in vitro reporter assay, the authors showed that the SVA_F element introduce a splicing acceptor site competing with the canonical site. Lastly, the authors showed that the SVA_F1 insertion is on a young haplotype that appears to be quickly rise in frequency, suggesting the insertion is under positive selection.

Overall, this manuscript presents a well-organized study. The writing is clear, and the datasets and analyses were described sufficiently. The authors provide a potential explanation of the dynamic change in pigmentation during the human evolution, through the regulation of ASIP expression. I only have a few specific comments.

We appreciate these positive comments about our work and the helpful suggestions below.

- **Page 4: the SVA_F1 subfamily was defined in the article “5-Transducing SVA retrotransposon groups spread efficiently throughout the human genome” (PMID 19652014). Please cite the reference when first mentioning SVA_F1.**

Thank you for bringing this reference to our attention. We have added the reference to that part of the manuscript on (p. 4).

- **Fig. 1f: It would be useful to label the index SNP rs6059655 in the plot.**

We agree and have added the rsID label to Fig 1f. In doing so, we noticed a minor error in the r^2 we had reported for rs6059655 (which was actually the r^2 for another SNP on the same haplotype) and so we have corrected this.

- **With the data described in the study, it is possible to estimate the age of the haplotype containing the SVA F1 insertion (see PMID 21816865 for an example). It can provide more insight into the time frame when the insertion occurred.**

We appreciate this helpful suggestion to use SNP-haplotype information to estimate the age of the SVA F1 insertion. We have now done so using the Relate software package, which can efficiently perform such analysis on hundreds or thousands of genome sequences (Speidel et al. 2019 *Nat Genet*; <https://www.nature.com/articles/s41588-019-0484-x>). Applying Relate to the 1KGP CEU and GBR populations produced age estimates of 14.3-25.4kya for the SVA F1 polymorphism. We now report these results in the main text (p. 8-9) and Methods (p. 38), and we have included a new Extended Data Figure 8 displaying the coalescent tree estimated by Relate at this locus (copied below for convenience).

Extended Data Figure 8. Genealogy of haplotypes at the ASIP SVA F₁ insertion.

Coalescent trees estimated by Relate for **(a)** CEU and **(b)** GBR haplotypes at the SVA F_1 insertion site. The purple branch contains all haplotypes carrying the SVA F_1 insertion. Age (in years) on the y-axis assumes a generation time of 28 years.

Reviewer #2:

This is a clearly written paper that describes fine mapping to identify a likely causal mutation that explains skin color variation and associated traits (i.e., skin cancer risk) among Europeans at the ASIP locus. Informatics analysis is given that argues that the causal mutation is a polymorphic SVA insertion, with a model where the polymorphic SVA insertion suppresses the creation of a nonfunctional transcript isoform through formation of an RNA hairpin with another SVA element that is present in all humans. This is consistent with prior knowledge on the potential of SVA insertions to cause splicing defects and leads to a model where an ancestral SVA disrupted ASIP function and that a second mutation found in some humans leads to a partial restoration. In other species, multiple types of mutations, including transposable elements, have been described at ASIP that play a role in coloration phenotypes.

This is a very nice effort to fine map a casual variant that would be further strengthened by additional functional data to test the model, including RNase protection assays to show that the proposed hairpin actually forms and more sophisticated minigene splicing assays to iteratively test effects of combinations of the two SVA elements. Such functional data would strengthen the paper, but I am not sure whether they would substantially enhance the broader significance of the finding.

We appreciate the careful review and positive comments about the work, and we appreciate these suggestions for ways to potentially expand the functional analyses. We undertook such experiments and despite managing to clone the fixed (shorter) SVA F element, we were unable to isolate a clone containing the longer, polymorphic SVA F₁. We suspect that the difficulty arises from challenges in amplifying the longer SVA F₁ (3.3kb, containing an expanded GC-rich VNTR) in contrast to the fixed SVA F (1.6kb). We tried several different standard high fidelity polymerases (Q5, Phusion, Kapa HiFi) with various concentrations of additives (betaine, DMSO) that have been reported to aid in amplification of GC-rich sequences but were still unsuccessful in cloning the polymorphic SVA F₁.

In light of these difficulties, we have opted not to further deepen the functional assays, in order to complete this revision in a reasonable time frame, as we agree that while further functional data would be interesting, it would not change the broader significance of the finding, and the main findings of the work are already well-supported by the existing functional data together with statistical fine-mapping and RNA-seq analyses.

We also note that the functional model we propose—in which inverted pairs of retrotransposons modulate splicing via formation of an RNA hairpin—does now seem to have conceptual support from recent work on inverted Alu elements at another locus – specifically, a preprint that found reverse-complement Alu pairs to mediate exon skipping in the *TBXT* gene (Xia et al. 2021; <https://www.biorxiv.org/content/10.1101/2021.09.14.460388v1>), and an analysis of splicing effects of

inverted Alu pairs across the human genome recently presented at ASHG 2023). We have added references to this work in the main text (p. 8).

The authors did an outstanding job in integrating genomics and biobank datasets to fine map the locus, but it is not clear to me that the lack of long read genomes and pangenome references are really the key resources for this type of study. For example, the SVA variant described here is relatively easy to discover and genotype since the most commonly used human reference genome has the ‘insertion allele’, making the variant appear as a “deletion” which does not require mobile-element specific informatics tools to analyze. In fact, this variant appears to have been first described in humans in 2007 (Korbel 2007 and Levy 2007, DGV accessions nsv435737 and essv4114058) and was found by every sequencing-based SV study since then that included Europeans, notably including the 1000 Genomes Project. Thus, it is a mystery why the statistical genetics community has failed to incorporate such SVs into their analysis; the authors here are the rare few in human genetics who are doing such.

Thank you very much for pointing this out and looking up these references. We agree and have now noted in the first paragraph of Results (p. 4) that the SVA variant was observed by Levy et al. 2007 and Korbel et al. 2007. We have also reworded the first paragraph of the Discussion (p. 9) to clarify that the effect of the SVA (rather than the SVA itself) had previously not been noticed, and we have removed the mention of pangenome reference panels, which we agree have not been the major bottleneck.

The tradeoff of decreased pigmentation and increasing body hair is a nice idea but would benefit from evidence in terms of the timing of the relevant mutations.

We appreciate this helpful suggestion and have now undertaken analyses to investigate the timing of both SVA insertions at *ASIP*.

To estimate the date of the recent, polymorphic SVA F₁ insertion, we used the Relate software package (Speidel et al. 2019 *Nat Genet*; <https://www.nature.com/articles/s41588-019-0484-x>), which efficiently infers haplotype genealogies from sequence data. We applied Relate to the 1KGP CEU and GBR populations and obtained estimates for the age of the polymorphic SVA F₁ ranging from approximately 14.3-25.4kya. We have added these results to the main text (p. 8-9) and Methods (p. 38) and included a new Extended Data Figure 8 displaying the coalescent tree estimated by Relate.

To roughly date the ancient, fixed *ASIP* SVA F, we compared its sequence to the sequences of other SVA F elements in the GRCh38 reference, which might 1) give clues about the historical order of retrotransposition events within the SVA F family and 2) allow estimation of insertion time based on the

numbers of observed base-pair differences within the Alu-like and SINE-R SVA sequence elements (chr20:34222672-34223023 and chr20:34223717-34224222 at *ASIP*) and the *de novo* mutation rate. We ultimately concluded that these sequence elements were insufficiently long to accumulate enough mutations to allow a precise date estimate, but we did discover that the most closely related SVA F (chr2:169785008-169787441, matching 349 of 352 bp in the Alu-like region and 505 of 506 bp in the SINE-R region) is polymorphic (based on entries in DGV and gnomAD-SV). This suggests that despite being fixed in modern humans and archaic hominins, the *ASIP* SVA F has probably been active relatively recently and may be a relatively young member of the SVA F family (which is ~3 million years old according to Wang et al. 2005: PMID 16288912). We have now noted this observation in the main text (p. 9) and provided details in Methods (p. 38).

We also conducted a literature review to better understand the state of knowledge regarding the broader timing of pigmentation changes and hair loss in humans. Broadly, the history of human pigmentation changes appears to have been complex, shaped (and reshaped) by varying evolutionary forces involving migrations to different environments and by changes in cultural practices (reviewed by Jablonski 2021, which we now cite). The timing and evolution of hair loss also appears to be uncertain; an analysis of increased constraint at *MC1R*—indicative of pressure to maintain eumelanin production—suggests that hair loss likely occurred at least 1.2 million years ago, but this estimate presupposed pigmentation constraint to be contingent on hair loss (Rogers et al. 2004; <https://www.journals.uchicago.edu/doi/abs/10.1086/381006>). In light of the remaining uncertainty around the timing of the SVA F insertion relative to loss of body hair in humans, we have removed the hair images from Fig. 5d, solely focusing the figure on changes in pigmentation.

Other minor comments to consider:

Line 106: if space allows further elaboration on the origin of the SVA subfamily may help, the mention of MAST1 exon is somewhat jarring to a casual reader without context.

We agree and have clarified that the *MAST2* exon 1 was 5'-transduced into the source element of the SVA F1 subfamily. We have also added a reference to Damert et al. 2009 ("5-Transducing SVA retrotransposon groups spread efficiently throughout the human genome"; PMID 19652014) for further context.

Line 134: should be than all OTHER SNP and indel variants

We have made this correction.

Line 139: why is tanning response the best-powered?

Comparing genome-wide associations for skin color (Fig. 2c) vs. tanning response (Extended Data Fig. 2a), the relative effects of different variants on skin color vs. tanning response vary quite substantially across top loci. Based on these data, we hypothesize that ASIP plays a larger role in pigmentation following sun exposure than at baseline.

Line 154+figures – any ideas for why Tibila Nerve also shows the expression association?

One hypothesis we have is that this could be due to melanocytes and Schwann cells (which ensheath peripheral nerves) sharing a common developmental origin. We have now noted this and added a relevant citation to Discussion (p. 10).

Line 157 Figure 3e colors do not seem to match – I do not see red for alternative 5' exons

Thank you for pointing out this error. We have corrected the colors listed in the legend.

Extended Data Figure 7c – the ddPCR design could benefit from more description to increase clarity for readers unfamiliar. What are the blue green and black circles, etc.?

We agree and have clarified the figure legend accordingly: “The arrows represent forward and reverse primers and the rectangles with circles represent quenched fluorescently-labeled probes, where the blue circle is a FAM fluorophore, the green circle is a HEX fluorophore, and the gray circles are quenchers that are cleaved during polymerase extension.”

Extended Data Figure 8. It is hard to see the IGV screen shots. They would benefit from higher resolution (no need to force 4 panels together in a supplementary figure) and by including the Repeat Masker track in the IGV view to visualize the location of SVA in the reference.

We agree with both points and have split the screenshots into Extended Data Figures 9 and 10 and added the RepeatMasker track (with detailed descriptions in the figure legends of how to locate the SVA breakpoints from the RepeatMasker annotations).

Methods: some inconsistency as to whether version numbers are listed for programs

We have now added version numbers throughout.

Line 720/749. It is not clear why different criteria were employed in the two data sets.

We have now clarified in Methods (p. 35) why we used different read-count criteria to genotype the polymorphic SVA in UKB vs. 1KGP: "Different linear separators were used here based on observed differences in the relative presence of n_{INS} and n_{ANC} for each genotype, presumably due to slight differences in sequencing and alignment parameters (e.g., average coverage, fragment length, bwa-mem options).

Line 937 – scripts available upon request is frowned upon these days. It is not hard to put anything that is relevant in GitHub/Zenodo to share.

As suggested, we have now uploaded the custom scripts we used for genotyping each cohort (1KGP, GTEx, UKB) as well as for calculating allelic fold change in Zenodo and have updated the Code availability section accordingly (p. 44).

Decision Letter, first revision:

28th February 2024

Dear Nolan,

Your revised manuscript "A sequence of SVA retrotransposon insertions in ASIP shaped human pigmentation" (NG-LE63191R) has been seen by the original referees. As you will see from their comments below, they find that the paper has improved in revision, and therefore we will be happy in principle to publish it in Nature Genetics as a Letter pending final revisions to satisfy the remaining requests and to comply with our editorial and formatting guidelines.

We are now performing detailed checks on your paper, and we will send you a checklist detailing our editorial and formatting requirements soon. Please do not upload the final materials or make any revisions until you receive this additional information from us.

Thank you again for your interest in Nature Genetics. Please do not hesitate to contact me if you have any questions.

Sincerely,

Kyle

Kyle Vogan, PhD
Senior Editor
Nature Genetics
<https://orcid.org/0000-0001-9565-9665>

Reviewer #1 (Remarks to the Author):

In the revision of the manuscript, the authors addressed all the comments, and the manuscript quality has improved.

The only minor issue is the authors stated that the codes for the project is available on Zenodo, but the record given cannot be found.

Reviewer #2 (Remarks to the Author):

The authors have responded to all of the concerns and the revised manuscript is improved. This is an interesting finding.

Final Decision Letter:

21st June 2024

Dear Nolan,

I am delighted to say that your manuscript "A sequence of SVA retrotransposon insertions in ASIP shaped human pigmentation" has been accepted for publication in an upcoming issue of Nature Genetics.

Due to the importance of these deadlines, we ask that you please let us know now whether you will be difficult to contact over the next month. If this is the case, we ask you provide us with the contact

information (email, phone and fax) of someone who will be able to check the proofs on your behalf, and who will be available to address any last-minute problems.

Your paper will be published online after we receive your corrections and will appear in print in the next available issue. You can find out your date of online publication by contacting the Nature Press Office (press@nature.com) after sending your e-proof corrections.

Before your paper is published online, we will be distributing a press release to news organizations worldwide, which may very well include details of your work. We are happy for your institution or funding agency to prepare its own press release, but it must mention the embargo date and Nature Genetics. Our Press Office may contact you closer to the time of publication, but if you or your Press Office have any enquiries in the meantime, please contact press@nature.com.

Please note that *Nature Genetics* is a Transformative Journal (TJ). Authors may publish their research with us through the traditional subscription access route or make their paper immediately open access through payment of an article-processing charge (APC). Authors will not be required to make a final decision about access to their article until it has been accepted. Find out more about Transformative Journals

Authors may need to take specific actions to achieve compliance with funder and institutional open access mandates. If your research is supported by a funder that requires immediate open access (e.g. according to Plan S principles) then you should select the gold OA route, and we will direct you to the compliant route where possible. For authors selecting the subscription publication route, the journal's standard licensing terms will need to be accepted, including <https://www.nature.com/nature-portfolio/editorial-policies/self-archiving-and-license-to-publish>. Those licensing terms will supersede any other terms that the author or any third party may assert apply to any version of the manuscript.

If you have not already done so, we strongly recommend that you upload the step-by-step protocols used in this manuscript to protocols.io. protocols.io is an open online resource that allows researchers to share their detailed experimental know-how. All uploaded protocols are made freely available and are assigned DOIs for ease of citation. Protocols can be linked to any publications in which they are used and will be linked to from your article. You can also establish a dedicated workspace to collect all your lab Protocols. By uploading your Protocols to protocols.io, you are enabling researchers to more readily reproduce or adapt the methodology you use, as well as increasing the visibility of your protocols and papers. Upload your Protocols at <https://protocols.io>. Further information can be found at <https://www.protocols.io/help/publish-articles>.

Sincerely,
Kyle

Kyle Vogan, PhD
Senior Editor
Nature Genetics
<https://orcid.org/0000-0001-9565-9665>